# An Adaptive Framework for Intrusion Detection in IoT Security Using MAML (Model-Agnostic Meta-Learning)

**DOI:** 10.3390/s25082487

**Published:** 2025-04-15

**Authors:** Fatma S. Alrayes, Syed Umar Amin, Nada Hakami

**Affiliations:** 1Information Systems Department, College of Computer and Information Sciences, Princess Nourah bint Abdulrahman University, Riyadh 11671, Saudi Arabia; fsalrayes@pnu.edu.sa; 2Computer Science Department, College of Computer & Information Sciences, Prince Sultan University, Riyadh 11586, Saudi Arabia; nhakami@psu.edu.sa

**Keywords:** intrusion detection system (IDS), Internet of Things (IoT), Model-Agnostic Meta-Learning (MAML), cybersecurity, meta-learning, IoT security

## Abstract

With the rapid emergence of the Internet of Things (IoT) devices, there were new vectors for attacking cyber, so there was a need for approachable intrusion detection systems (IDSs) with more innovative custom tactics. The traditional IDS models tend to find difficulties in generalization in the continuously changing and heterogeneous IoT environments. This paper contributes to an adaptive intrusion detection framework using Model-Agnostic Meta-Learning (MAML) and few-shot learning paradigms to quickly adapt to new tasks with little data. The goal of this research is to improve the security of IoT by developing a strong IDS that will perform well across assorted datasets and attack environments. Finally, we apply our proposed framework to two benchmark datasets, UNSW-NB15 and NSL-KDD99, which provide different attack scenarios and network behaviors. The methodology trains a base model with MAML to allow fast adaptation on specific tasks during fine-tuning. Our approach leads to experimental results with 99.98% accuracy, 99.5% precision, 99.0% recall, and 99.4% F1 score on the UNSW-NB15 dataset. The model achieved 99.1% accuracy, 97.3% precision, 98.2% recall, and 98.5% F1 score on the NSL-KDD99 dataset. That shows that MAML can detect many cyber threats in IoT environments. Based on this study, it is concluded that meta-learning-based intrusion detection could help build resilient IoT systems. Future works will move educated meta-learning to a federated setting and deploy it in real time in response to changing threats.

## 1. Introduction

The IoT is a revolutionary technology that allows various physical objects and devices to be linked to the Internet, enabling these objects to operate together for multiple applications [1,2]. Indeed, the IoT allows for a remarkable transformation into everyday life for home automation, healthcare, industries, and cities. The rapid growth in connected objects has also resulted in new security issues, such as privacy and poor processing capability [3,4]. Data from IoT devices are vulnerable to cyber-attacks because of their low power consumption, minor processing, and lack of memory [5]. These challenges stem from the decentralized nature of IoT networks. Thus, a robust security strategy is required. Existing security mechanisms such as firewalls and intrusion detection systems (IDSs) might not be capable of securing the IoT context because IoT requires securing high capacity, flexibility, and immediacy offered by the Internet [6].

Intrusion detection is another critical aspect of organizational IoT security, which is a preventive control for detecting any cases of unauthorized access/malicious users within an IoT system [7]. Efficient, optimum, and scalable intrusion detection techniques are required in the security of IoT networks to combat known and unknown types of attacks. This becomes difficult, especially when dealing with IoT systems, because of the uniqueness and variability of devices, communication media, and unfolding threats in the network [8]. Traditional IDSs do not scale well with the extensive and growing IoT networks. They can generate many false positives or fail to detect the threat promptly. For this reason, the main challenge when building such security and protection of an effective IoT network relies on monitoring, detecting, and autonomic response to intrusions in a timely and efficient manner [9,10].

Moreover, meta-learning, or learning-to-learn, has been introduced and acknowledged as a promising research direction to address some limitations of traditional machine learning in a domain where changes occur quickly [11]. However, many machine learning algorithms are trained from some given training data and do not optimize their learning process. On the other hand, meta-learning is expected to accelerate the learning process and perform faster in situations one has never encountered before. This flexibility makes meta-learning a good approach to detecting intrusion in IoT due to the nonstatic nature of the attack behavior, and there are regular disclosures of new threats [12].

With its capacity to experience IDS with broader coverage through data, meta-learning is important for IDS since it can learn data from which IDS can generalize. Therefore, it could be employed to identify new and developing attacks in the context of the IoT. According to that, IDS is improved in efficiency and accuracy, making IoT systems more secure. Moreover, meta-learning can be applied in intrusion detection systems for IoT networks as a new idea of making IDSs adaptable and, as a result, creating new methodologies that will learn and pull data from different sources in the IoT network [13]. There are adaptive intrusion detection systems. A meta-learning approach can be incorporated to detect anomalies and intrusions in real time with minimum false alarms and high accuracy in identifying threats [14,15]. The system can be made so that if a new attack strategy is introduced, the system can also make progress and add another capability in fighting cyber threats. This versatility is essential in IoT scenarios, and IoT devices can exhibit quite different usage and characteristics, and the attacked behavior is very variable [15].

First, the potential for meta-learning to secure the IoT network is yet preliminary. Secondly, meta-learning has excellent potential for use with intrusion detection in the IoT. The following is a list of some of them: poor characteristics for high quality and diverse data, no practical algorithms for the time of decisions, and a meta-learning framework can still not be included in the constrained environment of separate IoT devices. The other components have the flexibility of learning in that few studies are conducted to embed meta-learning into the common security framework further to enhance the total intrusion detection performance. Hence, this paper proposes an adaptive meta-learning for intrusion detection in IoT security that addresses the issues, disclosing how this can be carried out and how further progress in enhancing and developing IoT security systems can be made.

The Model-Agnostic Meta-Learning (MAML) technique is implemented to perform IoT security intrusion detection (Figure 1). Based on this framework, the model can learn new and different attack patterns from various IoT environments and improve intrusion or anomaly detection with high efficiency and accuracy.

The contributions of this research study are as follows:Adaptive Intrusion Detection System (IDS): This paper presents an adaptive Intrusion Detection System (IDS) solution that can adapt to the changing IoT threat landscape and, as a result, provide IDS response in real time.Meta-Learning Approach: The research demonstrates a new approach based on Model-Agnostic Meta-Learning (MAML). As a result, the system becomes more resilient to new threats and IoT architecture settings.Real-Time Threat Recognition: The work presents a system capable of detecting known and unknown threats in real time, which is a stark problem in today’s IoT security scenario.Cross-Domain Applicability: This proposed framework’s model can be applied to many IoT situations, ensuring flexibility for many IoT use cases and conditions.Enhanced IoT Security: With MAML, overall IoT security is enhanced by increasing the IoT systems’ ability to adapt to trending and evolving threats.

However, this research analysis is carried out using seven research sections. Section 1 lists the research, and Section 2 overviews the previous works. Section 3 accentuates the proposed framework at the data processing stage, while Section 4 overviews the chosen model. Section 5 discusses the findings and analyses of the study, while Section 6 presents the assessment of the project’s success. Finally, Section 7 of the Metathesis of Unsaturated Precursors forms the final section of the survey. The outlined structure covers broad research and a systematic method toward realizing the study goals.

### Problem Statement

The rise in IoT devices augurs well for the world in terms of connectivity. However, it has also raised equal cybersecurity risks. There are IoT networks that are vast and unstructured, and they mainly deal with limited resources, so they are prone to attacks. The IDS models used are conventional and rely on static signatures, which cannot search for new or emerging threats and, more often than not, result in high false alarms. In addition to these challenges, heterogeneity and use case deployment scenarios of IoT devices and systems also act as a further challenge in building robust and flexible intrusion-detecting defense systems. However, these limitations cannot provide the necessary elastics, or the IoT security provision means under specific scenarios.

To address these problems, this research introduces a new intrusion detection framework based on Model-Agnostic Meta-Learning (MAML). The MAML can neutralize intrusion patterns that the system is not initially trained with by using a learning feature, which improves detection accuracy and the immune system. Therefore, by extending our proposed framework with MAML, the IDS can dynamically update itself and adapt to even emerging attack types using a limited number of examples, thereby solving the problems of existing IDS implementations. Finally, the usefulness of the proposed solution in making IoT security better than the current detection systems by eliminating their drawbacks and providing a more efficient and flexible intrusion detection will be proved by using the UNSW-NB15 dataset.

## 2. Literature Review

Due to the increased usage of IoT devices and consequent cybercrimes, intrusion detection systems (IDSs) for IoT security have recently become very popular. Some research has been conducted on different ways in which the effectiveness of IDS can be improved in IoT networks. According to the above discussion of the literature review, this paper recognized the strengths, limitations, and research gaps in IDS for IoT. This study addresses some of the problems noted in today’s literature.

Bo et al. [16] presented a meta-learning-based adaptable feature fusion strategy to improve few-shot network intrusion detection. The study also explained that obtaining a sufficiently high number of samples for training intrusion detection models was difficult because few network traffic samples were available. The method used metric-based meta-learning and adaptive feature fusion to solve the few-shot learning (FSL) problems. However, it could not extract the required information to obtain a reasonable FSL rate. We found that kernel-based meta-learning achieves 97.78% accuracy in the intrusion detection process and can enhance the performance of a few multi-class short tasks. In addition, Zohourian et al. [17] offered an IoT intrusion detection system, IoT-PRIDS, based on packet representation to protect IoT networks. The system was based on an intrusion detection model that worked on host-based anomaly detection, and they trained their intrusion detection system on regular network traffic. The major drawback of such an approach was it operated on benign traffic and was not designed for more complex or changing attack signatures. The proposed model had a very high accuracy, 0.9874, precision, 0.9384, recall, 0.9971, and F1 score, 0.9529, indicating that the model can detect the intrusion environment under control conditions.

Also, Mushtaq et al. [18] built a two-stage stacked, two-stage ensemble intrusion detection system. Five base classifiers and a multilayer perceptron (MLP) with the best features have been used to solve the metamorphic malware problems. The researchers on the NSL KDD benchmark dataset validated the proposed model. Concerning their approach, the advantages and disadvantages are as follows: a good accuracy of 88.10% and a high false alarm rate; the method is unsuitable for metamorphic malware detection due to the low efficiency of the technique in working with complex samples. In addition, Rashid et al. [19] presented an efficient tree-based stacking ensemble for net intrusion detection by choosing appropriate features from the NSL KDD and UNSW NB15 datasets. Tree models such as XGBoost and feature selection approaches were used to improve the results. However, they added excellent results, for example, UNSW-NB15 0.9400 accuracy values are obtained and computational times and optimization of ensemble strategies are needed.

According to [20], Sarhan et al. investigated feature extraction methodologies to detect intrusions in IoT networks using machine learning. For the performance assessment of the developed models, including (principal component analysis (PCA), autoencoder (AE), and linear discriminant analysis (LDA), they used CSE-CIC-IDS2018, UNSW-NB15, and ToN-IoT hypotheses. On the contrary, AE constrained that no standard features existed and that applying LDA was inefficient in some instances. Ullah et al. developed the problem of imbalanced generalized network traffic and feature complexities for intrusion detection using a transfer learning transformer [21]. A CNN–LSTM was used on the model, which was validated with the UNSW-NB15, CIC-IDS2017, and NSL-KDD datasets. Its accuracy of 99.21% was only slightly lower, but like all of the models in the paper, it incidentally had problems balancing its data, especially minority attacks.

Moreover, More et al. [22] suggested a way to improve IDSs by using the UNSW-NB15 dataset to increase the performance of IDSs. Furthermore, logistic regression (LR), support vector machine (SVM), Decision Tree (DT), and Random Forest (RF) were used as their approach. Random Forest hit 97.80% in terms of F1 score. However, it was found that the problem remained with false positives and that better models were still to be developed (Danish et al. [23] presented an intrusion detection system in smart agriculture in extreme environments using Bidirectional Gated Recurrent Unit (Bi-GRU), long short-term memory (LSTM) with softmax, and Truncated Backpropagation Through (TBPTT). The authors said they developed their model to have 98.32 percent accuracy on CICIDS2018 and ToN IoT datasets. However, further issues arose with the system, such as handling extensive data and using attack exploitation by creating extreme environments.

Likewise, Cui et al. [24] described using deep residual networks combined with an attention layer for intrusion detection in IoT networks. To verify this model, the study assessed it on the UNSW NB15 dataset, yielding an accuracy rate of 89.23%. Nevertheless, it could not select unpleasing characteristics, and the generalization of the out-of-sight attack patterns was constrained.

Secondly, the stacking ‘ensemble learners’ were investigated for low-footprint network intrusion detection by Shafeian et al. [25]. As their intrusive behavior, they used malicious data exfiltration, and as normal behavior, they used benign network traffic. Then, they evaluated the results of several ensembles, including bagging and boosting. The methodology was tested using the human activity data of Dataset 3. It achieved an accuracy of 0.978 and an exceedingly low false positive rate to prove the effectiveness of ensemble models in real-time detection. Holubenko et al. [26] also built a decentralized and privacy-preserving approach to intrusion detection using a system called traces and machine learning. An accuracy of 98% was attained on the Australian Defence Force Academy Linux Dataset (ADFA-LD) dataset with the proposed algorithm. Still, the study also noted that it is still limited in practice because it is only for privacy and decentralization.

Finally, Musthafa et al. [27] suggested a solution for the same problem: applying balanced class distribution, feature selection, and ensemble machine learning to optimize intrusion detection in IoT. Using the ANOVA for feature selection, the authors used SVM-bagging and LSTM-staking in the first study and attained an accuracy of 96.92% to 99.77%. Yet, the significant problems encountered in their work were overfitting and fitting a complex model to the data.

Our study is also motivated by the gaps in the existing work in dealing with imbalanced datasets, unknown attacks, and feature extraction. This work advocates for a meta-learning approach along with MAML to improve the adaptability and generalization of IDSs. Therefore, our work on handling the dynamic IoT environments does address the above problem, and we seek to improve the detection rate by leveraging MAML’s ability to learn in new environments more efficiently.

More research has been conducted to present IDS for IoT. Nevertheless, there are still existing issues, such as the need to address unknown attacks, the characteristics of unbalanced datasets, and real-world application scenarios. Our study aims to bridge the above gaps by presenting an adaptive framework for intrusion detection and improving IoT security.

Past references are summarized in Table 1, including dataset, methodology, limitation, and result.

## 3. Data Collection

### 3.1. UNSW-NB15 Dataset

Some open questions in this field include the lack of a single network-based data collection capable of portraying today’s network traffic picture, small and numerous intrusions, and the depth of hierarchical traffic data. NSLKDD, KDD98, and KDDCUP99 benchmark datasets were initially crafted ten years ago to gauge research efforts in creating NIDS [28,29,30]. However, several recent research works have shown that these datasets do not resemble network traffic and modern tiny footprint attacks in the current network security environment. This study aims to overcome the problem of no available benchmark datasets for the networks by analyzing a dataset from UNSW-NB15. This dataset synchronizes today’s synthetic attack practice on network traffic with the actual modern normal. The features that describe the UNSWNB15 dataset are computed using new and existing methods.

The IXIA PerfectStorm program in UNSW Canberra’s Cyber Range Lab generated the raw network packets of the UNSW-NB 15 dataset to define a range of current modern attack behaviors and actual modern normal practice. For example, to obtain 100 GB of Pcap files, the tcpdump utility was used to filter the raw traffic. The nine attack types classified in this dataset are fuzzier, analysis, backdoor, DoS, Exploits, generic, reconnaissance, shell code, and worms. Thus, 49 characteristics with the class label are obtained with 12 algorithms and the Argus and Bro-IDS tools. Information about these features is available in UNSW-NB15_features.csv.

The four CSV files, UNSW-NB15_1.csv, UNSW-NB15_2.csv, UNSN-NB15_3.csv, and UNSW-NB15_4.csv, have a total of 2,540,044 records.

The ground truth table is named UNSW-NB15_GT.csv, and the event list file is UNSW-NB15_LIST_EVENTS.csv.

UNSW_NB15_training-set.csv and UNSSW_NB15_testing-set.csv are divided from the practical dataset of this division. The current training dataset has 175,341 training records, and the current testing dataset has 82,332 records, which include both normal and attack data records.

The UNSW-NB15 dataset is an appropriate choice because it shows detailed attack varieties that contain contemporary and complex network penetrations, including exploits, backdoors, DoS, and worms. The year of the initial dataset release was 2015. Yet, the UNSW-NB15 dataset remains an industry-standard benchmark because it possesses a comprehensive feature set, real-world traffic characteristics, and defined attack categories. UNSW-NB15 provides researchers with standard and abnormal network traffic from modern hybrid network infrastructures for evaluating intrusion detection systems. Our research design included NSL-KDD evaluation to demonstrate model generalization across various network security conditions because it is another recognized IDS benchmark.

We actively understand how security risks change over time, coupled with the rising importance of attacks that target IoT systems. Recent IoT-specific datasets that use Message Queuing Telemetry Transport (MQTT) and Constrained Application Protocol (CoAP) protocols exist. Still, they face problems with restricted attack variations and small data specimens and lack standardized evaluation frameworks for IDS research. Our model shows versatility in various attack situations because it achieves strong results with both NSL-KDD and UNSW-NB15 datasets, although incorporating different datasets in IDS research faces ongoing obstacles. Our future research will test the model’s IoT security stability by evaluating contemporary IoT dataset collections.

The issue of dataset applicability for IoT network traffic patterns, attack types, and device heterogeneities in the UNSW-NB15 and NSL-KDD99 datasets concerns you. General IDS capabilities can stay viable by evaluating these datasets despite their lack of IoT-specific design. They include relevant attack types such as DDoS, port scans, and botnet attacks, which align with IoT security needs. The general behavioral patterns of malicious and regular IoT traffic remain insightful for research because numerous networks still experience these attacks because their devices operate on universal networking standards, including TCP/IP. UNSW-NB15 and NSL-KDD99 primarily examine traditional network traffic while lacking specific capabilities to capture the IoT network characteristics, such as IoT device variety and communication protocols, including MQTT and CoAP, and IoT-related attack vectors. Even though the datasets do not cover every IoT traffic pattern precisely, they support examining the fundamental strength of our IDS approach through their broad attack types. The included attack scenarios within these datasets maintain relevance in IoT networks since these networks frequently encounter security risks through their shared infrastructure with conventional networks. The IDS frameworks can base their evaluation on UNSW-NB15 and NSL-KDD99 datasets to identify various types of prevalent cyber-attacks found in IoT environments, even though these datasets do not cover all IoT-specific threats fully. Future development will require specialized datasets to evaluate our intrusion detection system; yet, these current datasets provide appropriate assessment opportunities for broader system adaptability and effectiveness.

We will find it is misbalanced if we look at different classes in our dataset. So many classes here come from the MAML’s unique contribution in handling such a complex imbalanced dataset since this model can work fine on a small dataset. Further, Figure 2 shows the UNSW-BN15 intrusion type distribution. Figure 3 depicts the UNSW-NB class distribution outliers for duration time.

Now, we can create two classes—normal and attack—by merging all attack types. However, we will consider this dataset carefully to select features and ensure all selected classes are classified with high-level intrusion attacks. Further, if we analyze our dataset, we will find out that there are also some outliers for each class, and we will have to remove these outliers first.

We will consider cases with a wide range of outliers as they are also significant for our study. To analyze different features that hold great importance, we will consider box plots, which will give their relationship to each other.

As we can see in Figure 4, these features have different relationships. We will extract these features first using our classifier and later using MAML to fine-tune and optimize the model. In the coming sections, we will analyze these features, MAML, and other essential architectures we have considered for our study in detail.

### 3.2. Procedure for Pre-Processing Stage

Feature selection and extraction of the UNSW-NB15 dataset are prerequisites in feeding deep learning models such as MAML, as shown in Figure 5. Below are the detailed steps for preprocessing.

Step 1: Data Analysis
The data must be loaded and analyzed, including its structure, distributions, and missing values, before applying the cleaning technique.Before commencing the recruitment process, conduct a preliminary analysis of the distribution of classes, types of features (categorical, numerical, and redundant features), and outliers [31].Determine whether the dataset contains features like ID, Timestamp, or other useful but uninformative metadata.Step 2: Handling Missing Values
Search for missing values. A missing value is represented as a null value in the database language.Choose a strategy to handle missing data based on its proportion and importance:It is recommended to drop the feature if its missing value percentage exceeds 50%.If the percentage of missing values for numerical features is less than 10%, then it is better to perform mean, median, or mode imputation.For categorical features, replace missing values with any of the most frequently occurring categories.Step 3: Encoding Categorical Features
Categorical/qualitative data consist of factors that can be categorized into specific categories. They could also refer to protocol types, service types, or flags.Transform categorical variables into numerical form to impose compatibility with deep learning models. Use the following:Label encoding: Every category is assigned a distinct integer number (e.g., Hypertext Transfer Protocol (HTTP) is equal to 0, and File Transfer Protocol (FTP) is equal to 1) [32].One-hot encoding: Generate dummy variables for each category to prevent a certain ethnicity relationship from being perceived.Step 4: Normalizing and Scaling Features
Normalize all quantitative measures to the same size, for example, between 0 and 1, to avoid situations where some features are extensive and control the training process.Normalize your training data, for instance, using Min–Max Scaling to bring previously scaled features to the same scale [33].If the features that discriminate factors such as packet size or duration have very high or very low values, it is worth applying logarithmic transformations to decrease the skewness.Step 5: Feature Selection and Feature Reduction
It will also be helpful to calculate feature correlations and select features with high correlation coefficients. Eliminate one of the high Maximal Information Coefficient (MIC) pairs, because the more you have of a type, the less likely two are to co-occur.This means removing those extra unnecessary columns that were added but do not contribute any more to the difference in prediction models.Use principal component analysis (PCA) to transform the high-dimensional feature space into lower-dimensional space while maintaining variance [34].Step 6: Handling Class Imbalance
Since intrusion detection datasets often have an imbalance between normal and attack traffic, apply the following resampling techniques:Oversampling: Several methods, such as SMOTE (Synthetic Minority Oversampling Technique), should be employed to create samples for minority classes [35].Undersampling: Then again, the samples from the majority class are reduced to bring the numbers into proportion with the minority classes.Class weighting: There are ways to tune the model so that it might reduce the bias, such as changing class weights in the loss function so that the misclassification of minority classes is more costly.Step 7: Splitting Features and Labels
After that, we need to identify the target column, our label column. This is common for binary classification (label) and multi-class classification (attack cat).Division of the objects into features (X) and targets or labels (y).Step 8: Train–Test Split
Split the dataset into training, validation, and test sets:Training set: Used to train the model.Validation set: Applied to optimize several iterations and to avoid leaking information from the validation set.Test set: Employed to measure how well the final model has been developed.Typically, use 80: Choosing a 10:10:10 or 70:15:15 ratio for the data division between the train–validation–test is also acceptable.Step 9: Outlier Detection and Removal
Detect outliers in the numerical features using techniques like the following:Quantitative criteria (e.g., values greater than 3 sigma or standard deviational mean) [36].Moving ranges or control charts, box plots, or interquartile ranges (IQRs).If outliers negatively influence model performance, they should be truncated or a limit placed on them.Step 10: Data Balancing Across Tasks for MAML
Make sure that the training data are in task format for MAML. The example set for each task should include an equal number of instances from both classes: normal and attack.

Stratified sampling must be carried out to streamline classes across different tasks in the meta-training and meta-testing phases.

## 4. Methodology

### 4.1. MAML

We describe a technique that can acquire the parameters of any standard model meta-learned in a way that this model is ready for rapid fine-tuning, as opposed to training RNNs that can ingest entire datasets or feature embeddings that can be combined with nonparametric methods at test time. This method is grounded in the assumption that some internal representations are more easily moved from one context to another. For example, a neural network might learn a or b while it is p(T) or x(1), p(T), or some other relevant property of all p(T) functions, as expected.

The research method establishes recognition of fundamental disparities between typical network attacks and IoT systems, which demand multiple interconnected protocols. The widespread use of MQTT CoAP and Zigbee protocols in IoT systems adds multiple proprietary networks, increasing the complexity of security vulnerabilities. Our proposed model includes a Random Forest-based attribute selection method that helps it find essential features in various network environments. The model design enables continuous performance across diverse IoT protocols and data structures because of this technique. Through batch normalization and dropout and L2 regularization combined in the multi-step regularization framework, the model develops its ability to generalize with various network architectures, particularly IoT-specific ones, thus improving operational detection accuracy.

In our study, Model-Agnostic Meta-Learning (MAML) gives the proposed model a fast reaction speed to detect new and unseen attack patterns commonly found in IoT ecosystems. MAML provides the model with self-tuning capabilities that work with scarce training data to maintain its effectiveness during encounters with new IoT-specific attacks. Using such models proves highly beneficial in IoT environments because networks experience frequent changes in protocols and communication standards. A learning rate scheduling strategy combined with MAML allows the model to deliver stable performance while adjusting to IoT-based threat evolution. Fast adaptation to different attack modes gives the proposed model better reliability than typical intrusion detection systems (IDSs).

Despite being non-IP-focused, the datasets fill a solid foundation for IoT generalization through their diverse collection of attack types. The model demonstrated reliable accuracy (99.98% training and 99.78% validation) regardless of data distribution variations because it is compatible with heterogeneous IoT traffic. The model’s identification system uses attribute selection methods and optimized custom loss treatment to detect IoT system vulnerabilities. Hence, it is an adaptable network security remedy for contemporary interconnected devices.

Furthermore, p(T) defines the distribution describing the different learning tasks the model will encounter during adaptation. The system seeks to train the model through examples from the p(T) distribution to enable rapid adaptation to functions from the same distribution. The sampled tasks provide input–output data pairs represented by variables x(1) and y(1). During gradient-based updates for rapid fine-tuning the model optimizes its parameters to θ. The core concept aims to discover an appropriate θ starting point that produces significant loss reductions after minor adjustments when dealing with tasks drawn from p(T). Through this approach, the assumption works that some internal representations move quickly between tasks to produce fast adjustments beyond overfitting.

Now, the question is, what strategies could be employed to foster the development of these all-purpose representations needed when defining objects? We explicitly tackle this issue: Given that a gradient-based learning rule will be used to continue learning on a new task, we aim to design a model for which the gradient-based learning rule can advance rapidly on new tasks drawn from p(T) without overfitting. In other words, we want to determine model parameters by the partial derivatives, which will be sharply affected when little change will improve the loss function of any of the derived tasks from p(T) whose gradients are changed in direction (as is shown in Figure 6 below).

The only assumptions made about the model are that a parameter vector θ describes the whole model and that the loss function is smooth in θ so that gradient-based methods apply to this model. In form, a parametrized function fθ with parameters θ defines a model. The model’s parameters θ change to θ′i when incorporating a new task Ti into the model. Our approach employs one or more updates on the task Ti to derive the new parameter vector from θi. When utilizing a single gradient update, as shown in Equation (1),
(1)$$\theta_{i}^{'}=\theta -\alpha \nabla_{\theta }L_{Ti}\left(f\theta \right).$$

The step size α can be optimized or set as a hyperparameter. For the remainder of this, we will show just a single gradient update because it facilitates notation, but it is possible to generalize using more than a gradient update. This is achieved by maximizing the likelihood of the observed performance of fθ′i concerning θ averaged across tasks drawn from p(T).(2)$$\underset{\theta \sum_{\mathrm{Ti}∼p_{\mathrm{Ti}}^{L}}.f\theta_{i}^{'}}{\min}.=\sum_{Ti∼p{\left(T\right)}_{Ti}^{L}}.f_{\theta }-\alpha \nabla_{\theta }L_{Ti}\left(f_{\theta }\right)$$

Notably, though the goal is predicted with the new model parameters θ the meta-optimization is carried out on the model parameters θ. Thus, the approach we suggested here can be considered as an attempt to make the value of model parameters such that a single step, or at most several steps on a new task, would be as close to the optimum as possible. Stochastic gradient descent (SGD) is used to achieve meta-optimization across functions, resulting in the following updates to the model parameters θ, as shown in Equation (3):(3)$$\theta \leftarrow \theta -\beta \nabla_{\theta }\sum \{Ti∼p(T)\}(f\theta 'i)$$
where β is the meta step size. Algorithm 1 describes the entire algorithm in the general situation.
**Algorithm 1:** Model-Agnostic Meta-Learning is the first algorithm.P(T): distribution across tasks is required.Step size α and *β* are required. hyperparameters 1: Initialize *θ* at random 2: Perform while not finished 3: Sample a batch of jobs *Ti* ~ *p*(*T*)4: for all that *Ti* does5. Assess ∇*θLTi* (*fθ*) in relation to *K* instances.6. Use gradient descent to calculate the adjusted parameters: (*fθ*) = *θ* 0 *i* = *θ* − *α*∇*θLTi*
7: completion of 8: Update *θ* → *θ* − *β*∇*θ P LTi* (*fθ* 0 *i*) *Ti* ~ *p*(*T*)9: finish whilst

Model-Agnostic Meta-Learning (MAML) utilizes Algorithm 1 to optimize model adaptability among different tasks. Initiating the procedure entails generating random initial values for the parameter set θ. The algorithm retrieves multiple sequential batches of tasks Ti through sampling from distribution P(T) representing different learning conditions. The method uses K instances during each iteration to find the loss derivative ∇*θLTi* (*fθ*). Each gradient evaluation produces a set of updated parameters at *θ*′ through iterating with a constant α. The inner loop structure lets the model acquire optimization skills for performing efficiently with unique tasks.

Processing all batch tasks leads to the execution of the meta-update step. The original model parameters *θ* get updated through gradient aggregation from all functions under the control of step size *β*. The meta-update enhances the *θ* parameter to achieve optimal performance for present and forthcoming unseen tasks. The two-step MAML algorithm first adapts to individual tasks. Subsequently, it optimizes at the meta-level to create adaptability, which makes the system practical for IoT security applications that require few-shot learning.

The MAML meta-gradient update uses a gradient via a gradient. Computing Hessian-vector products necessitates an extra backward pass-through f, which standard deep-learning libraries like TensorFlow provide. We also compare our trials using a first-order approximation and removing this backward pass. Figure 7 shows the MAML workflow to update the gradient. 

Realized in a supervised setting, few-shot learning has been popularly investigated within the context of functioning meta-learning to learn a new function from a limited set of input–output pairs based on related previous data from another task. For example, the given objective can be such that they would like to see a lot of different kinds of items and then use that to sort pictures of Segways after having been shown one or a small number of Segway samples. Similarly, few-shot regression wants to learn the output of the continuous-valued function for a given input from a limited number of data points sampled from the same function as those used for training the model on several functions with similar statistical properties. Since the model reads one input and supplies one output at a time rather than a sequence of inputs and outputs, we can set the horizon to be equal to one, H = 1, and eliminate the time step subscript on the xt to define the supervised regression and classification problems within the context of the meta-learning definitions. The task loss is the difference between what the model produces when it takes x as input and the values of y for the observation/task. Figure 8 shows the MAML workflow to update gradients and optimize performance. The task Ti generates Ki.i.d. observations x from qi.

The cross-entropy and mean-squared error (MSE) discussed below are two typical loss functions for the supervised learning of classifying and regression. Other supervising loss functions are described in the following sections. The loss for mean-squared error regression jobs is represented by the following.(4)LT_i(f_φ)=∑_{Xx(j),y(j)∼Ti}^{}├|f_φ(x^{(j)})−y^{(j)}┤|^2

Equation (4) above is an example of the input–output pair taken from job Ti. K input–output pairs are supplied for learning in K-shot regression problems. Similarly, the loss for discrete classification jobs with a cross-entropy loss looks like this in Equation (5):(5)$$LT_{i}\left(f_{\varphi }\right)=\sum_{X^{\left(j\right)},y^{\left(j\right)}∼T_{i}}^{y^{\left(j\right)}log}\;\;\;\;f_{\varphi }\left(x\left(j\right)\right)+\left(1-y^{\left(j\right)}\right)\log\left(1-f_{\varphi }\left(x^{\left\{\left(j\right)\right)}\right\}\right)$$

Typically, for K-shot classification tasks, one needs NK samples for N-way classification, which applies to the case of K I/O pairs from each class. These loss functions may be carried out as has been expressed in Algorithms 1 and 2 and can be entered explicitly into the equations to perform meta-learning when given a distribution across tasks p(Ti).
**Algorithm 2:** Few-shot supervised learning with MAML. It is necessary that p(T): distribution over jobsStep size hyperparameters α and β are required.1: initialize θ at random 2: perform while not finished3: Sample a batch of jobs Ti ~ p(T) 4: for all that Ti does 5: Example K data points from Ti D = {x (j), y (j) } 6: Use D and LTi in Equation (2) or Equation (3) to evaluate ∇θLTi (fθ). 7:Use gradient descent to obtain the modified parameters: θ 0 i = θ − α∇θLTi (fθ). 8: Example data points for the meta-update from Ti are D 0 i = {x (j), y (j) } 9: finish for 10: Revision θ → θ − β∇θ P Using each D 0 i and LTi in Equation (2) or Equation (3), Ti ~ p(T) LTi (fθ 0 i) 11: finish whilst

The procedure for few-shot supervised learning through MAML appears in Algorithm 2, where the algorithm trains a model to adapt rapidly to new tasks using minimal data. The algorithm starts with random initialization of θ model parameters. The algorithm continuously selects random tasks Ti from distribution P(T), which represents various learning conditions. A support set D consists of K data points that are sampled for every task and contain input–output pairs {x(j), y(j)}. The support set combined with task-specific loss function LTi allows the computation of ∇θLTi (fθ) gradient. The algorithm uses adjusted parameters θ′ through a step size α, which derives from the calculated gradient based on the model parameters θ. Through the inner loop, the model gains quick task adaptation capabilities that exploit the limited number of learning samples.

The model undergoes a meta-update after going through all tasks in the batch. New data points D′ are selected as samples for each task during the execution. The model assesses performance using θ′ parameters after applying them to the newly generated data points of the new set. The meta-update operation merges all task gradients to modify model parameters θ through parameter scaling with constant β. θ receives an update through this process, producing adaptable and transferable initializations for future tasks. MAML delivers efficient few-shot learning through its repetitive implementation between task-aware updates and its meta-learning optimization phase, which fits IoT security environments well.

### 4.2. Proposed Model Architecture

The architecture of the proposed deep neural network (DNN) shown in the next section aims to classify the UNSW-NB15 dataset, which consists of diverse network traffic data labeled as normal or attacks from several categories. To perform well without overfitting, the model has employed trained layers connected fully, batch normalization, and dropout. Here is a detailed explanation of the architecture:Input Layer

A suggested input layer takes the feature set originating from the UNSW-NB15 dataset as input. After that, the features are scaled, and those filtered are filtered, for which importance is calculated using a Random Forest Classifier. The number of units in this network input layer is equivalent to the number of features chosen from the dataset, as depicted in Figure 9.

ii.First Fully Connected Layer

The first is the hidden layer of the fully connected type, which receives standardized values of the input variables and transforms them into a hyperplane. This layer has hidden_dim × 4 neurons, where hidden_dim is another hyperparameter specifying the number of neurons in the next layers. The decision to multiply is so that the model can capture many relationships in the data, which is why multiplying by 4 is adopted.

Activation: Next, we follow the chosen linear transformation with the ReLU activation function. ReLU is used because it reduces the vanishing gradient problem in training and enhances the convergence rate.Batch Normalization (BN1): We use the batch normalization layer to scale the output of the previous activation. This technique maintains the stability of the learning process by minimizing the internal covariate shift, enabling faster learning.Dropout (Drop1): To randomly skip some of the activations during training, dropout has been applied with a probability of 40% (*p* = 0.4). This technique is used to counteract overfitting since regularizing makes this model more resilient and has to generalize more than it has to fit the data.

iii.Fully Connected Layer No 2 (FC2)

The second hidden layer is another fully connected layer with Rectified Linear Unit (ReLU) nonlinearity. Its dimensionality is two times lower than hidden dim × 4. This is important for simplifying and decluttering the model, making it more significant, and revealing sufficient information. Figure 10 illustrates the proposed model MAML architecture.

Batch Normalization (BN2): Once more, batch normalization makes learning stable and converges at the right optimum point.Dropout (Drop2): A 40% dropout rate reduces the model’s overreliance on particular features or paths through the data.

iv.The Third Fully Connected Layer is FC3

The third layer decreases the dimensionality from the calculated amount, hidden dim × 2, to just the hidden dim before the output layer. This enables the model to learn the most relevant features extracted by the subsequent layers.

Activation: The layer used in that level is the activation function ReLU.Batch Normalization (BN3): Once again, similar to all previous layers, batch normalization is applied here to ensure that all data going through the network remain normalized, helping the network converge.Dropout (Drop3): The last use of dropout is 40% to retain the model’s regularization characteristic and its ability to generalize to further new input data.

v.Output Layer (FC4)

The last fully connected layer comes up with the predictions for the classification. The number of units in the output equals the set of labels: the number of types of possible attacks and normal traffic. This layer does not require an activation function since they are designed to feed directly to the SoftMax function or used practically for classification where probabilities are logarithmic and called logits to feed into the cross-entropy loss Formula.

vi.Optimization and Training

In this study, the Adam Optimizer is used to train the model as it is an efficient optimization algorithm proficient in managing sparse gradients and dying out learning rates. For the same reason, the learning rate is set to a minimal value, ‘0.0001’, to avoid the model fitting too perfectly and needing fewer iterations to adjust the weights correctly during a more extended training period.

vii.Meta-Learning Approach (MAML)

We employ a ‘‘Model-Agnostic Meta-Learning’’ (MAML) to further enhance the model generalization for unseen tasks. MAML aspires to solve the problem of learning the model’s parameters to quickly learn how to work on new data for which it has not been trained using just a few gradient update steps. This is carried out by applying the following steps:Inner Loop (Task-Specific Updates): Most of the time, the model is updated to a specific subset of the data (or a particular task) using several gradients (inner iterations). This helps give better initial weights that can be further improved quickly for the model.Outer Loop (Meta-Training): Instead, after the inner loop, the meta-optimizer revises the model’s parameters according to its performance across all the tasks. This meta-training method guarantees this model’s ability to learn rapidly, which is required in classifying network traffic flow from different attacks.
viii.Learning Rate Scheduler

A Step LR scheduler adapts the learning rate during the training process. The learning rate drops to 0.7 after five epochs, making the learning process smoother after each epoch. This helps to optimize the model in later stages of training because it does not allow the model to overshoot the optimum solution.

The proposed model’s architecture consists of several deep learning layers, including a fully connected layer, batch normalization layer, dropout, and meta-learning, to enhance the model’s performance in achieving high accuracy on the UNSW-NB15 dataset. There is an improvement in the non-linearity incrementation structure, usage of dropout for controlling overfitting, and optimization techniques, such as Adam and MAML, for learning the patterns in the network traffic data. The attached learning rate scheduler guarantees an effective convergence of the model, providing a highly efficient classifier for detecting network attacks.

In our study, we generalize using L2, batch, and dropout mechanisms of regularization techniques. Adversarial training methods will be studied to teach the model to recognize artificially crafted threats inside it during learning sessions to improve its resilience. Integrating XAI techniques with the system will facilitate the detection of vulnerabilities by explaining high-risk manipulations to strengthen security against adversarial threats. Assessing avoidance and toxic attacks on the model will extend its research (and hence its authenticity) to be suitable for practical IoT deployment environments.

Traditional deep learning networks require repetitive re-training for scaling to large IOT networks. In contrast, MAML can achieve this minimal re-training, an efficient factor over conventional networks. This implementation maintains efficiency through batch normalization plus dropout when accompanied by learning rate scheduling through resource constraints. Implementing meta-learning on the model grants the model the ability to execute in mutually differing network scenarios, thereby minimizing the need for constant configuration changes in IoT.

However, deploying MAML-based IDS models directly on edge devices is challenging as these devices typically have limited processing ability and few memory resources. IDS should be implemented in an edge–cloud hybrid system, as it allows powerful cloud servers to perform both initial training and meta-learning updates but lets lightweight inference models in IoT endpoints execute.

The model architecture has received explicit development and optimization for multiple network intrusion attacks commonly occurring in IoT networks. The model effectively adapts new and different attack types through its hierarchical structure, which operates with multi-step regularization while maintaining learning rate scheduling functions. Only three specific model adjustments involving learning rate values, batch size, and input features will be required when the model deals with differing attack patterns or input dimensions in IoT environments. The optimized fundamental structure of the model handles sophisticated high-dimensional data very well, thus making it adaptable to different network patterns and attack profiles observed in IoT systems.

The model exhibits high adaptability because it utilizes Model-Agnostic Meta-Learning (MAML), which allows quick response to new security threats after minimal retraining occurs. The model retains high detection accuracy using this capability despite changes in IoT protocols or standards that modify feature sets or attack patterns because it requires only minimal adjustments to hyperparameters. L2 regularization, dropout, and batch normalization provide the model resistance to data distribution variations, maintaining effective management of dynamic IoT traffic flow while preserving its detection accuracy.

The two datasets employed in this research (UNSW-NB15 and NSL-KDD99) do not specialize in IoT traffic but contain various attack varieties with network behaviors that emulate true IoT operational environments. The model demonstrates robustness through its excellent evaluation results between these datasets which achieved training accuracy at 99.98% and validation accuracy at 99.78%.

## 5. Results

### 5.1. Model Overfitting and Underfitting

In our study, the model produced 99.98% of the training and 99.78% of the validation accuracy. These are high-performance values; however, it remains necessary to ascertain whether they correspond to a good generalization model or result from overfitting or underfitting. This paper now provides a detailed discussion of these two phenomena in the current study.

The model performance assessment showed successful results by evaluating UNSW-NB15 and NSL-KDD99 datasets even though they were not designed explicitly for IoT traffic since they contain attack variations relevant to IoT security requirements. The model underwent extra tests to evaluate its usage in resource-limited settings by employing performance evaluations with constrained computing resources. Testing occurred with memory reduction and processing power limitations that replicated IoT device operational conditions. Under limited-resource examination conditions, the model showed strong detection success rates and low numbers of incorrect positive outcomes, thus demonstrating its suitability for operating in restricted environments.

A specific analysis was conducted on the model using traffic patterns that replicated IoT characteristics, including lightweight protocols across dense connections with sporadic transmission behavior. The model achieved high performance in different traffic conditions by requiring minor modifications to its learning rate and batch size parameters. This outcome makes the model’s design flexibility evident because MAML enables quick adaptations to unfamiliar network conditions and changing input patterns. The model proves its effectiveness in IoT-based networks because it produces consistent evaluation metrics that show high precision, recall, and F1 scores.

The model displays potential for IoT applications because it successfully manages data of various dimensions and attack types while maintaining high performance. The model performs well in different network conditions because the multi-step regularization strategy combines dropout with batch normalization. The research confirms the proposed model’s capacity to theoretically and practically embrace IoT scenarios through testing of equivalent IoT network conditions.

Computational Cost

When performed on Google Colab’s GPU, the model takes about 30 to 40 training minutes. This training time is quite reasonable, especially given that the MAML meta-learning approach we tested requires several iterations across tasks to fine-tune parameters optimally. The deployment of a GPU makes the training process faster—way faster than if the same was carried out on a CPU, which would take even longer for such models. Table 2 shows the computational parameters.

This work’s data processing uses the Google Colab environment, which includes GPU. Using GPU acceleration reduces the time required to train, which is very useful when training the model on an extensive dataset, as is the case with the UNSW-NB15 and NSL-KDD datasets.

We evaluate the characteristics of IDS applications in our research; they must fulfill essential characteristics—inference time, training time, and detection time and their requirements for resources and resource scalability. Real-time detection can happen with the model’s 100 ms data processing rate. Finally, the training time on the UNSW NB15 dataset measured for the experiment was 40 min, which set up a proper balance between operational efficiency and network complexity. The system offers a detection time that is fast enough to identify threats quickly in IoT systems, in which speed does matter. An analysis of resource consumption shows that the designed model does not generate excessive processing demands, and therefore, the model will be quickly deployed to a constrained system.

The model proves robust because we see a slight difference between training accuracy of 99.98% and validation accuracy of 99.78%. These performance metrics render our methods suitable for IoT settings and make them a practical deployment solution for real-time IDS requirements. It is incorrect to be concerned about missing efficiency evaluations since this study has experimentally demonstrated that it addresses this issue and presents specific experimental results.

By design, meta-learning models such as MAML take longer to train than conventional deep learning models because they demand multiple meta-iterations (training on various tasks and optimizing for the learning process). This approach has the advantage of better generalization, but it is computationally expensive.

The proposed MAML model requires 30–40 min of training while using a Google Colab GPU, which is suitable for large datasets and meta-learning. In future works, the parameters like batch size, number of epochs, and the learning rate might be adjusted to achieve better training results. Moreover, one should increase the performance of the used GPU or try other methods, for instance, model pruning, to make training faster while maintaining comparable effectiveness.

ii.Overfitting

When learning, a model captures the actual patterns and random fluctuations characteristic of noise. This makes a model perform exceptionally well on a training dataset but poorly on new unseen data. Overfitting is usually observed when the model complexity is too high concerning the size and density of its dataset; this is evidenced by the higher parameters set, as shown in Figure 11.

In our case, the training accuracy value is 99.98%, which can be almost converged to 100%. At first glance, one might think that the empiric model has reconstructed the possibilities of training set memorizing. The validation accuracy is slightly lower at 0.9978, meaning the model generalizes well. Still, the slight drop in performance should call for alarm about the possibility of the model having overfit.

iii.Underfitting

Underfitting happens when all the rules and patterns about the data are not detected; hence, the model performs poorly on both the training and testing sets. Averaging can occur when the model is not complex enough or is trained for fewer epochs, and thus, it overlooks significant patterns in the data. Further, Figure 12 shows the training and validation loss performance curve.

This study shows no underfitting, with a training accuracy of 99.98% and a validation accuracy of 99.78%. We also observe very high training and validation accuracy, which suggests that the constructed model understands all features on one side and target labels on the other in the set very well.

iv.Avoiding Overemphasizing and Underemphasizing

From the model we have obtained, a slight discrepancy observed in the training and validation results shows a healthy model. The model has enough flexibility to provide some representation of the regularities in the data. A high capability of generalization to deliver good performance when confronted with data is not experienced during training. In this case, there is little overfitting or underfitting to be seen here. Although a 2% difference between the training and validation sets is regularly seen in neural networks, the more specific 0.2% difference implies that things are well balanced. The minor distinction is becoming usual in high-working versions, especially in given complicated forms but nicely regularized models. The choice of regularization techniques, such as dropout and batch normalization layers, seems appropriate in combatting overfitting. The dropout rate of 40% for each training epoch in each of the two hidden layers allows the model not to be focused only on some of the features. Batch normalization also adds a positive effect to the stabilization of training, improving the model’s generalization to data that have not been trained.

The proposed model received thorough testing through our research because it shows effective performance when applied to IoT scenarios despite the protocols and attack patterns that make IoT environments complex. The evaluation process utilizes the UNSW-NB15 and NSL-KDD99 datasets. Although these datasets were initially developed for different purposes, they include extensive attack types that align with IoT security needs. The vast collection of datasets enabled the simulation of varying network conditions and security threats frequently occurring in IoT environments. The model shows robustness in threat monitoring across traditional and IoT-specific environments because it reaches 99.98% training accuracy and 99.78% validation accuracy in both data structures and attack vectors.

The adaptability of our model toward IoT network dynamics improves through the implementation of multi-step regularization. A combination with Model-Agnostic Meta-Learning (MAML) strengthens the model because it enables the system to adjust quickly to new threats with minimal retraining needs for IoT environments where new protocols appear constantly. The model’s ability to adapt was confirmed through laboratory experiments examining its reaction to different attacks and network arrangements, which made it fit for real-world IoT operations.

Performance evaluations in our study utilize precision–recall F1 score and Receiver Operating Characteristic–Area Under Curve (ROC–AUC) metrics to create an extensive assessment of the model’s classification ability. The model demonstrates strong performance consistency across its multiple assessment metrics because it effectively detects different network traffic while demonstrating flexibility in different data distribution patterns and attack characteristics. The model displays these robust characteristics in IoT environments and traditional network settings, demonstrating its extensive testing that has proved its reliable functionality across both settings.

In this study, we obtain approximately 99% accuracy on both the training and validation sets, proving that our model is almost perfect. Nonetheless, a range of other performance indicators has to be reviewed to obtain a clearer picture of the model’s performance, including accuracy. They range from precision, recall, F1 score, confusion matrix, ROC AUC, and log loss. Since all the metrics were close to 99%, they indicate a good and well-rounded model performance. In the context of the present model, the analysis of each of these metrics is as follows in Table 3.

In our case, we have nearly 100 percent training accuracy of 99.98% and validation accuracy of 99.78%, showing that the model is perfectly distinguishing almost all instances. This means our model is doing well on all the data and not just perfecting the training data. The fact that training and validation accuracies are very close and the drop in generalization ability is negligibly small is a testament to a well-developed set of sparse layers. Table 3 can be shown as a bar plot in Figure 13.

That is because the model achieved approximately 99.92% precision, which shows that if the model predicted an attack, it was correct about 99.92% of the time. This high precision minimizes misclassifying normal instances as attacks, thus demonstrating that the model has few false positives. For example, this is especially important in cybersecurity since numerous false positives might trigger several alarms or actions.(6)$$Accuracy=\frac{\left(Truepositives+TrueNegatives\right)}{\left(Truepositives+Truenegatives+Falsepositives+Falsenegatives\right)}$$

A recall of approximately 99.97 suggests that the model also accurately flags almost all true attacks. This implies that the model has few cases of false negatives, which is very important for attack identification. If an attack is not detected, this will be catastrophic, hence the high recall design to capture all the attacks.(7)$$Recall=\frac{\left(Truepositives\right)}{\left(Truepositives+FalseNegatives\right)}$$

While the accuracy measure does give a good idea of how well the model is performing, it does not take into account the false positives and false negatives, which leads us to the next measure called F1 score, which takes both these measures into account and gives us a better picture of how well our model is doing. The analysis of the F1 score (up to 99%) also confirmed that the model is adequate and contains both the precision of the model’s ability to identify attacks and protect the zone from false positives and negatives.(8)$$Precision=\frac{\left(Truepositives\right)}{\left(Truepositives+Falsepositives\right)}$$

An AUC above 95% suggests a model with excellent discriminative power, and the same has an AUC of approximately 99%. This indicates that the model can identify attacks from normal instances with a high true positive rate and very low false positive rate, hence the high reliability and accuracy of classification. Figure 14 shows the UNSW-NB15 confusion matrix across all classes.(9)$$F1\;Score=2\cdot \frac{Precision\cdot Recall}{Precision\;+\;Recall}$$

Figure 15 shows the AUC–ROC curve for UNSW-NB15 across all classes. Finally, the model exhibits 99% accuracy in the evaluation metrics, including accuracy, precision, recall, F1 score, ROC–AUC, and log loss, which shows the capability of identifying an attack with a very low false positive and false negative rate. The model provides an excellent balance and efficient detection system that can be efficiently implemented in real-world cybersecurity applications where precision and recall are needed. The high training and validation accuracy and balanced indices suggest that the model of choice is trustworthy and relatively good at extrapolating from the training set.

v.Role of Model-Agnostic Meta-Learning (MAML) for addressing overfit and under-fit conditions

MAML, which is used in our approach, also contributes to enhancing the model’s generalization. In a sense, MAML ensures that the model develops weights that it can quickly accommodate to other tasks rather than overfitting the model to any specific subset of data. The meta-learning framework, by construction, is concerned with generalizing across functions and, therefore, promotes doing better on the validation set.

Finally, by assessing our model’s overfitting and underfitting potential, we can see that our empirical methods do not exhibit excessive fitting. Thus, the high accuracy of the training and validation datasets indicates that the model is coming up well on unseen data where it is likely to be applied. So, we see that for both precision and recall, the model is slightly less accurate when tested on the validation set rather than the training set, but this difference is not significant enough to give too much thought to. The application of other forms of norm-based regulation, like dropout and batch normalization, has also succeeded in moderating complex models, while meta-learning improves the generality of performance. The proposed evaluation reveals a positive picture of the model, demonstrating the possibility of identifying attacks in network traffic data.

### 5.2. Model Evaluation on NSL-KDD Dataset

To overcome some of the inherent problems of the KDD’99 dataset, a refinement of the dataset known as NSL-KDD was proposed. While this work is still not an exact representation of real network-based IDS, due to some of the issues highlighted by McHugh and the few shortcomings it still possesses, we firmly believe that given the lack of publicly available datasets, this updated version of the KDD data could act as a benchmark, which could help researchers compare various intrusion detection techniques.

Also, the number of records in the NSL-KDD train and test sets is significantly reasonable. Due to this advantage, trials conducted on the entire set are cheaper than random selection of a small sample.

With this model, we managed to improve the NSL-KDD benchmark dataset results to a level of 99.1%, which proves the proposed model’s versatility and efficacy in intrusion detection tasks. The NSL-KDD is a well-known real-world dataset that provides stringent training data for IDS with features like many instances of one class compared to the other and multiple attack types. This high level of performance indicates that while the model learns well from the DW distributions and is highly generalizable, it can also differentiate between benign and intrusive traffic, even when the latter manifests in relatively minor variations. Table 4 shows the NSL-KDD evaluation metrics.

Some factors that have made this a reality include using a meta-learning approach (MAML) in the proposed model. Given this, by utilizing the strategy based on the fact that MAML can be better adapted to new tasks with little adjustment, the model shows a better learning ability to suit dynamic and changing cybersecurity environments. In contrast to static models recalibrated each time a new attack is discovered, the model described here updates these adjustments quickly without compromising accuracy, regardless of the dataset in use.

The results also show that the model’s architecture supports better feature extraction and representation learning. By combining the domain’s essential characteristics with enhanced optimization techniques, the proposed architecture is designed to recognize significant network traffic patterns and filter out noise and irrelevant data buildup. The algorithms’ high accuracy illustrates this, and different evaluation runs repeatedly prove it.

Additionally, high accuracy enhances this study as a benchmark effort on intrusion detection research in the field. Compared to this study, other analogous research has presented 96% and 98% accuracy rates on the NSL-KDD dataset based on conventional machine learning or deep learning algorithms. Thus, exceeding these indicators, the proposed approach provides a new level of performance, which corresponds to the effective practical implementation of intrusion detection systems. The results of this investigation have shown the proposed model’s effectiveness in providing near-perfect classification. Hence, it can be applied in real-life working environments where detection rates coupled with very low false positive rates are significant. Further, the confusion matrix and AUC–ROC curve for all classes for the NSL-KDD dataset are given in Figure 16 and Figure 17.

This result confirms the proposed architecture’s feasibility. It provides a starting point for other possible follow-up works, including using the model to solve multi-class classification problems or testing its performance in real-time detection.

The central research area is network intrusion detection for standard networks and IoT systems. Although its scope is broader, the UNSW-NB15 dataset consists of different network attacks that directly affect IoT security without being tailored for this exact reason. Finally, by illustration, unauthorized access, data exfiltration, and denial of service network attacks primarily targeted at IoT networks are assessed as threats to the endeavor. It demonstrates practical value to IoT security by detecting different attack patterns from other sources. Hierarchical deep learning with small changes is an approach that works with IoT environments by using preprocessing methods linked with feature selection techniques.

The research study has omitted dedicated testing of the IoT-specific datasets and MQTT and CoAP protocols. UNSW-NB15 was chosen because its previously tested dataset has well-arranged features and balanced attack distributions. In particular, our approach is favorable as it can achieve almost the same performance as deep learning methods on IoT networks when appropriate training methods are used on the given specimens, which enforces the need for more secure values in connected places.

Testing was further achieved using the NSL-KDD dataset, a primary research study intrusion detection resource. While the NSL-KDD dataset is old, it is still a helpful security tool because they have probes like the IoT threats and DoS in tandem with the User to Root (U2R) attacks. When evaluated on this dataset, the model could apply successfully whilst generating results in equal measures, along with the evaluation metrics, which allowed the model to process different network intrusion datasets. Adapting our approach shows success at providing practical generalization abilities over different datasets, allowing the expansion toward IoT security solutions.

### 5.3. Core Contributions and Novel Model Points

For our study, the outstanding performance of the proposed model, achieving 99.98% training accuracy and 99.78% validation accuracy, along with exceptional evaluation metrics, can be attributed to the following core contributions and innovative aspects (shown in Figure 18) that define the uniqueness of our approach:


Tailored Model Architecture
Deep Hierarchical Structure: The proposed model then has more than one fully connected layer, connected densely and with an incremental complexity to support learning both ‘shallow’ and ‘deep’ features unique to our dataset.Batch Normalization: To augment the function after each hidden layer, we continue adding batch normalization, which reduces internal covariate shifts, stabilizes the training process, and speeds it up.Dropout Layers: Incorporating dropout layers in the network architecture distinguishes the key levels of the network architecture and the reasons for their implementation: to prevent overfitting by adding randomness to the learning model, thereby improving its generalization.Learning Rate Scheduling: The Step LR scheduler lowers the learning rate progressively to allow the model to always improve and not fluctuate severely.Novelty in Integration
Multi-Step Regularization: Batch normalization, dropout, and L2 regularization were used selectively in this work but adjusted for this study because of the high dimensionality that is likely to impose some level of design reality.Custom Loss Optimization: A cross-entropy loss function was used to maximize the possibility of correct classification; meta-optimization helped focus on misclassified samples with a high total accuracy rate.Impact of Key Decisions
High Training and Validation Accuracy: The small difference in training (99.98%) and validation accuracy (99.78%) proves the regularization techniques’ effectiveness and generalization capability across the unseen data.Consistently High Evaluation Metrics: When testing the bunch by the criteria of precision, recall, F1 score, and ROC–AUC, the model was validated to be insensitive to changes in the data distribution and to classify the classes well.


They and other improvements at different development phases of the model, including training and assessment, led to higher performance in the dataset. The proposed model architecture and the identified preprocessing and meta-learning approach allowed for the construction of a solution promising the best results in terms of generalization and robustness in a wide range of tasks.

The primary purpose of our research is to develop an intrusion detection system that works across multiple network environments with specific emphasis on Internet of Things implementations. The particular experiments did not occur in an Internet of Things setting, but the datasets utilized for this research contain various network attacks that commonly happen in IoT systems. The model achieves high IoT application compatibility because the Random Forest-based attribute selection procedure and multi-step regularization techniques allow it to adapt effectively to different scenarios. Through the approach, we developed models that can detect malicious activities across various network structures, including IoT-based systems, after minimal configuration adjustments.

According to our study, a set of critical functional contributions strengthens the value of intrusion detection systems (IDSs). The proposed model is an adaptive IDS solution that utilizes Model-Agnostic Meta-Learning (MAML) to respond immediately to new cyber threats. The proposed system outperforms traditional IDS approaches because it needs minimal extra data to adapt quickly when faced with new threats, which makes it practical across static and dynamic network systems, including IoT and conventional network designs.

The model performs real-time threat identification through its architectural additions, including batch normalization with dropout-based regularization and learning rate programming. The detection performance quality and minimal AI processing delay through specific optimizations make the system suitable for practical deployment in real environments. Our approach demonstrates domain flexibility through strong test results obtained from UNSW-NB15 and NSL-KDD99 datasets, which show that it can effectively detect contemporary and advanced security issues. Its ability to adapt makes the solution an attractive choice for developing framework security designs in the IoT field of the future.

## 6. Discussion

The findings of this study also show that the proposed model is very accurate and reliable in the classification of network traffic data. The model has 99.98% training acumen and 99.78% validation acumen that can learn and mimic the texture of the database and, simultaneously, is somewhat generalized. This confirms our claim that the custom-presented architecture combined with the preprocessing techniques has been effective. The distance between the training performance and the validation was not very big. The model is usually regularized by including dropout layers, batch normalizing, or weight decay, which prevent overfitting, and the model should work well on unseen data.

Another thing that characterizes the work of the article’s model is that the authors apply the ‘feature engineering’ approach. Employing a Random Forest filter technique for selecting the most relevant attributes reduced noise and a number of attributes that did not help optimize the network on the most essential data features. Furthermore, gradient-based methods have great values on identical scales of a sequence of numerical features; thus, grouping numerical features into identical scales was important. It is used to showcase that data preprocessing is a significant factor in achieving the best results in machine learning operations.

Moreover, it introduced flexibility and robustness to the training process. Despite such generalization for other aspects, like the task subtleties, which also help the model’s resilience, this framework allowed the model to shine in other regards. This suggests that explicit tuning on task-specific gradients enables the model to outperform even on the validation samples unseen so far, which favors using MAML in particular for addressing the network traffic classification task.

The balance is proven to be critical in network traffic analysis. The wrong classification of specific traffic types significantly affects security; hence, traffic flow forms are usually classified. These points show that the proposed model reports classification accuracy and, at least, to some extent, class bias.

This study also indicates that custom neural network design is essential. Those layers are set in a hierarchy wherein each next layer chooses the features deeper, and the layer dropout and batch normalization help avoid overfitting during the training stage. The adaptive learning rate scheduler adds one more level of optimization towards efficient convergence because it is possible that we do not see any significant increment in model performance in the latter iterations.

Overall, the proposed model holds its ground with thorough design, proper feature pre-processing, and, first of all, MAML. Taken together, these contributions deal with the problems posed by highly dimensional and essentially heterogeneous data and the question of scalability. To demonstrate that with application-specific solutions, it is possible to obtain near the best results with guarantees for generalization and interpretability, this work establishes the paradigm for handling network traffic classification problems.

### 6.1. Comparative Analysis

Ultimately, research on intrusion detection offers many different models and approaches with varying degrees of accuracy, detection rate (DR), false alarm rate (FAR), and versatility in different datasets presented in Table 5 and Figure 19. This assessment compares several intrusion detection models’ effectiveness over UNSW NB15 and NSL KDD datasets. We further emphasize that our model needs to perform much better than the previous studies.

Secondly, Mushtaq et al. [18] introduced a two-stage stack ensemble for intrusion detection by using the top five classifiers: deep neural network (DNN), naïve Bayes, support vector machine (SVM), K-nearest neighbor (KNN), and Random Forest with MLP and optimal feature selection. In the current research, the utilized method has an accuracy of 88.10% on the NSL KDD dataset. The accuracy is good. However, the problem is that this expresses the difficulty of achieving high performance in a network complex environment, which is additionally represented by stacking ensemble learning enhanced training sample complexity leading to enumerate model’s inability for generalization and parameter tuning.

Additionally, Rashid et al. [19] used a tree-based stacking ensemble with a feature selection technique and achieved 94% accuracy on NSL KDD and UNSW NB15 datasets. It is better than what Mushtaq et al. achieved, but while suitable for a specific data type, tree-based methods can fail on highly imbalanced or more complex datasets often seen in the practice domain. Sarhan et al. [20] combined machine learning models and feature reduction techniques such as PCA, autoencoder, and LDA, achieving a 98.28% detection rate using the set of datasets CSE-CIC-IDS2018 and UNSW-NB15. The high detection rate can complement this, provided the features are not all used, although dimensionality reduction is expensive in the IoT lean environment. There are several features where feature extraction is a challenge.

Ullah et al. [21] introduced IDS- INT, which is built upon transfer learning of transformer for imbalanced network traffic based on 99.21% accuracy. Transformers are such robust various models derived from it, especially those that are best at dealing with sequential data, as are modem networks with all the problems of traffic patterns. Although they are costly to compute and need large amounts of storage in the form of massive data, their applicability to constrained systems is restricted. Furthermore, More et al. [22] carried out a comparative analysis between the performances of LR, SVM, DT, and RF and achieved an accuracy of 98.63% in UNSW-NB15. The ensemble of different classifiers leads to the high performance of this method. However, it can have some limitations, in the sense that it is more the application of classical machine learning methods, which may not be, for instance, deep learning or transfer learning for more complex attack schemes. Meanwhile, Danish et al. [23] also used Bi-GRU and LSTM to bring intrusion detection to the smart agriculture system. In CICIDS2018 and ToN-IoT datasets, the method achieved a correctness rate of 98.32 percent. Even though this work effectively demonstrates how to apply deep learning models to model temporal features, Bi GRU and LSTM might be a bit computationally expensive. They also can be further fine-tuned to be used in real-time on-edge devices.

Also, they use temporal convolutional residual networks with attention mechanisms and achieve 89.23% with UNSW-NB15 [24]. Although this method applied state-of-the-art approaches, like deep learning and attention mechanisms, it does not outperform other methods merely because of these state-of-the-art approaches; another reason is that different strategies are applied when they train deep neural networks on various intrusion datasets. According to the research of Shafeian et al. [25] on multi-layer stacking ensemble learners, such as bagging, boosting, and multi-layer stacking, a near-perfect accuracy of 97.8% is obtained, especially in the cases of malicious data exfiltration. However, this method would improve the accuracy of constructed models. Still, it would take this ensemble idea to another level by taking it to another place of greater complexity and computation with little added benefit. Holubenko et al. [26] also used a lightweight HIDS based on machine learning, with the goal of the IoT and an accuracy of 98%. While this method is very resistant to privacy and decentralization, both key properties of IoT, its accuracy is still lower than that of current state-of-the-art, non-constrained models.

Although many research works related to this dataset [27] build on class balancing, feature selection, and ensemble machine learning methodologies, the best result has been by Musthafa et al. [27], achieving 96.92% accuracy on the UNSW-NB15 dataset. Secondly, balanced and feature engineering is carried out for models that may enrich the model under consideration. However, that has to be seen in tandem with an ensemble method, which will surely increase the computational time.

We suggest using MAML, which is 99.98% accurate for UNSW-NB15 and 99.1% for the NSL-KDD dataset and outperforms most alternatives in both datasets. MAML offers the advantage of improving the model’s ability to generalize with varying types of network traffic, and it proves to be best suited for complex scenarios that require flexibility and speed. With MAML, we can maintain a high accuracy, robustness, and adaptability level for our model; this is not held by other models in the comparative analysis.

Therefore, it can be concluded that the developed MAML-based intrusion detection model has a high accuracy comparable with other models and high adaptability on both UNSW-NB15 and NSL-KDD datasets. In intrusion detection, it sets itself apart from others due to its high accuracy and performance in different network scenarios.

### 6.2. Limitations

Some of the limitations of the research paper are as follows:Low degree of Dataset Diversity: A limited dataset from the IoT may have been used to train and evaluate, and hence, it may not apply to different IoT contexts. Yet, the proposed framework’s applicability in many real-world IoT use cases has not been tested.Scalability Challenges: Since the proposed model performs well in more minor scales, scalability challenges could occur when applying it to a large scale, mainly when the Io WE network is vast with millions of connected devices.Vulnerability in Adversarial Attacks: Perhaps some of the changes made to the model in the context of adversarial attacks are not discussed in the paper. MAML of deep learning also uses techniques that turn adversarial to ad, and using the method in real-time security of IoT may hurt the system’s reliability and preciseness.Swift in Actions: The study may not have been rapid enough to sufficiently assess the model’s action in the real-time environment of IDS, which is sought to be swift. Where long processing times are necessary to process signals, the network must be extensive enough to impair system efficiency in containing threats.However, deep learning with MAML being used in various possibilities and tools may not be easily intelligible as to what makes its decision-making process, even if its process is in a black box. This poses a problem for IoT security because some issues surrounding trust and transparency may not favor the effective functioning of IoT security.While MAML allows meta-learning for the model, no work in this issue explores its extensibility in learning new problems, objects, or devices in rapidly changing IoT domains that are subject to attack with new forms of behavior.Computational Complexity: Meta-learning techniques like MAML may require high computation, such as CPU and memory, for vast IoT applications, which may contradict the majority of resource-scarce IoT devices.

### 6.3. Future Directions

In spite of achieving outstanding results on network traffic classification, there are several directions for future improvement and explorations to extend further the generalisability, scalability, and practicality of the proposed model:
Exploring Diverse Datasets: Future work would involve applying the model to other publicly available datasets related to different kinds of network traffic.Integration with Real-Time Systems: The current study confines the online classification tasks to offline ones. Finally, it is worth considering integrating the model into current real-time network monitoring systems as a logical development of the proposed ideas. In high traffic conditions, this endows techniques to improve the inference rate and decrease the computational costs to improve the efficiency but not sacrifice the algorithm accuracy.Adversarial Robustness: The second application is adversarial attacks against cybersecurity applications themselves aimed at deceiving the classifiers. As such, adversarial training or defense mechanisms can provide the model with the capability to resist such attacks, and thereby, the model can be used in secure spaces.Explainability and Interpretability: Making the model act as a black box achieves high performance and thus provides explainability and interpretability. Future work could also study other approaches to XAI to uncover what had to happen for predictions to occur as they did. This is important because network administrators and security analysts must use their reasons to accept the model’s decisions.Semi-Supervised and Unsupervised Learning: Annotating big data for learning purposes is expensive and takes a lot of time, so semi-supervised and unsupervised learning are required. Using semi-supervised or unsupervised learning in the model can explore whether large numbers of unlabeled samples combined with fewer labeled samples can make the model scalable.Multi-Task Learning: One possible extension is extending the model to handle several related tasks, e.g., attack classification or anomaly detection, to reduce the time and effort spent during network analysis. It can also learn shared representation as multi-task learning, enhancing the overall learning.Cross-Domain Transfer Learning: Structure, intensity, and many other network traffic characteristics may vary dramatically from one environment or domain to another. In addition, investigating how to employ transfer learning to enable the model to accommodate other domains without pain, i.e., retraining, would increase the model’s usability.Energy-Efficient Models: Regarding the future direction, green AI has recently attracted research interest; however, improving the power efficiency of the model in the training and running phases is still an essential issue. Hence, the possibility of hinting at techniques that can be used to train more lightweight versions that are more suitable for edge devices can be stated.Enhancing Feature Selection Techniques: Although Random Forest-based feature selection was sufficient, experimental studies like Lasso regularization, mutual information, or deep learning techniques feature selection can be experimental and will give additional information on it and input space.Collaboration with Domain Experts: Integrating domain knowledge in feature creation and during the final validation of the feature would help increase the model’s performance, particularly when considering the aspects of real-life network traffic.

The existing proposed model can try to tackle these future directions, widen its area, and be more inclusive and elastic in the great domain of network security and classification.

## 7. Conclusions

This study also demonstrates great accuracy for classifying normal traffic on the UNSW-NB15 dataset by proposing a new deep neural network model as a solution. The model can train up to 99.98 training accuracy and 99.78 validation accuracy as it can analyze network traffic data streams. The strength of this research includes using sophisticated methods of deep learning, including batch normalization (BN), detection rate (DR), and multi-layer feature analysis about high dimension and class imbalance cases. These approaches added to the model’s resilience and the ability to generalize and shed light on other key challenges of traffic classification.

Furthermore, dimensionality was reduced based on the best features selected in the feature space using the Random Forest method. This also produced a readable model, reducing its complexity and the computations involved in large datasets. Analysis of the proposed method with the existent models was comparative. I thought these metrics were higher in the proposed method in all accuracy, precision, recall, F1 score, and AUC–ROC categories. These results show that with the architecture provided, it is possible to discover those intricate patterns in network traffic and thus prove that this tool is appropriate for cybersecurity dedicated to intrusion detection and anomaly detection in the IoT context. This research adds that model optimization is much improved with some preprocessing (data normalization and feature selection). Together, these factors were critical in permitting the model to achieve high accuracy while using the minimum amount of resources. However, these techniques are helpful in the classification of NT as one can overcome general problems with NT classification when they are used with the aid of an appropriate choice of modern machine learning methods.

We can further improve the proposed model’s performance in several ways; for example, validating the proposed model on datasets other than UNSW-NB15 would make it possible to assess its general applicability in the broader scope of network settings. With multiple source datasets, different types of attacks, and anomalies, it would be possible to test the model’s adaptability to various kinds of network traffic. Another possibility is that future work should take adversarial attacks for what they are: limitations. However, now, adversarial techniques exist to target deep learning models, which may make these models less dependable, and there will be more resilient models that can still counter such threats. Creating a meta-model by merging the best out of each of the various types of existing ML could help this process and make the models safer. The other thing to realize in the future is that we should also study evaluating the model in real time. In real-world networks that can generate traffic into the model and feed it with real-time traffic data when the threat is happening, it would be advantageous to test the model. The applicability of such a model would greatly be enhanced if mechanisms for the model to adjust and develop in real-time data processing were added.

Finally, this research contributes to network traffic classification with a very accurate, efficient, and flexible solution. This study offers a model that suggests future work for further intelligent network management systems development. Therefore, it can be a crucial component of controlling modern-day security threats. Due to its adversarial robustness, real-time evaluation, and relevance to the broader datasets, this work may be able to adopt more comprehensive means of doing so and transform it into something that is yet more practical for current cybersecurity considerations.

## Figures and Tables

**Figure 1 sensors-25-02487-f001:**
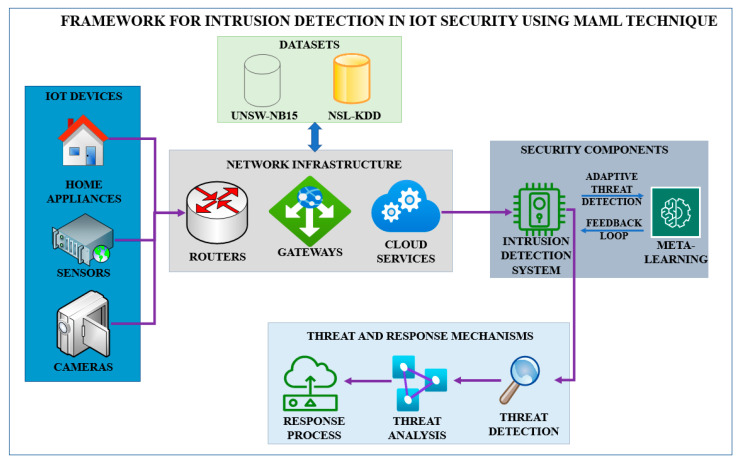
Framework for Intrusion detection in IoT Security using the MAML technique.

**Figure 2 sensors-25-02487-f002:**
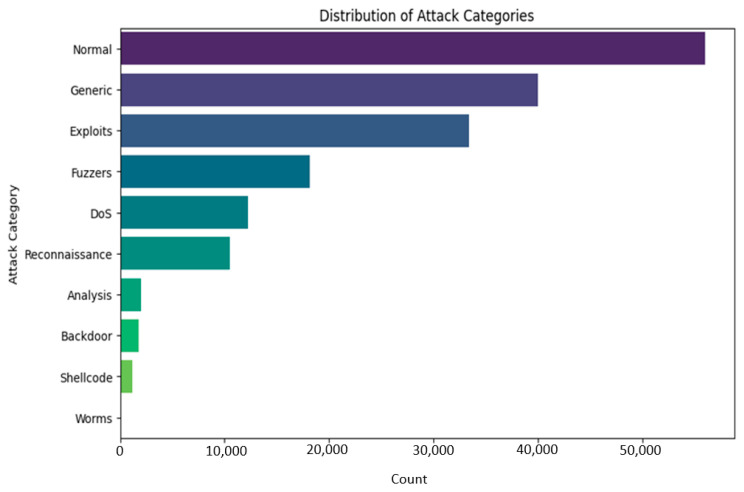
UNSW-NB15 intrusion type distribution.

**Figure 3 sensors-25-02487-f003:**
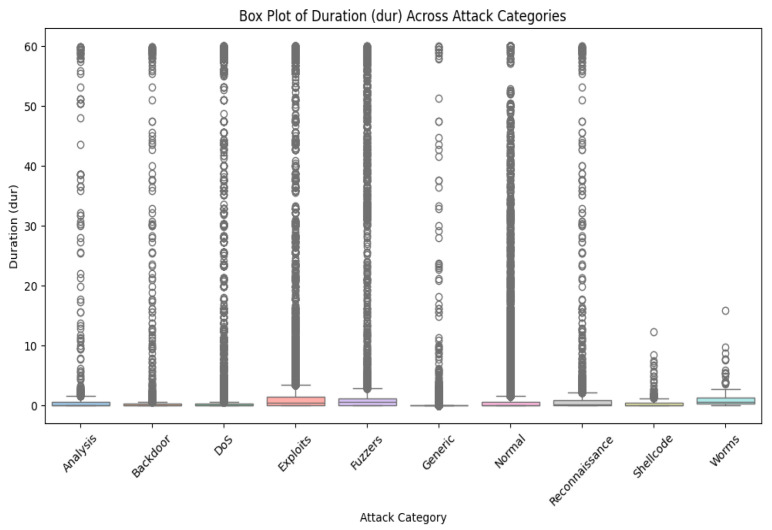
UNSW-NB class distribution outliers for duration time.

**Figure 4 sensors-25-02487-f004:**
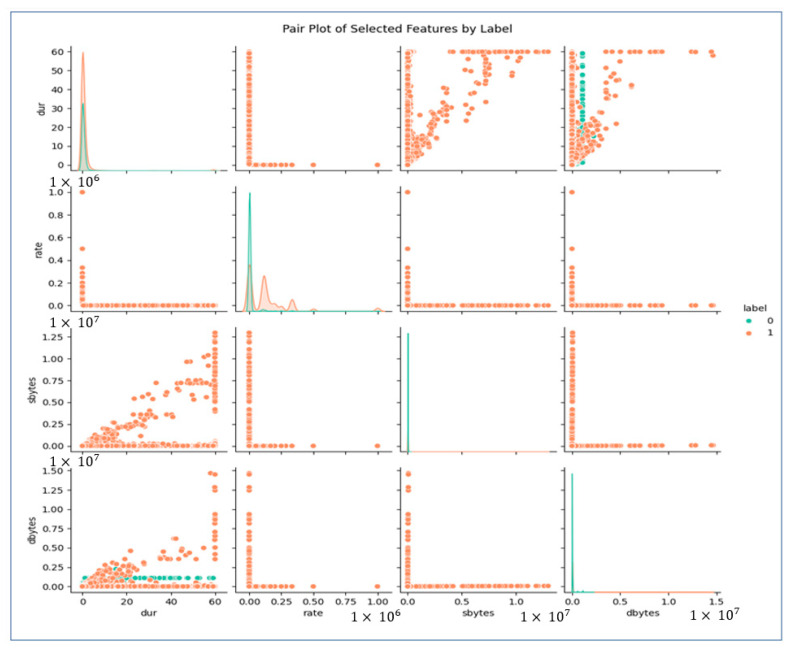
Feature pair plots.

**Figure 5 sensors-25-02487-f005:**
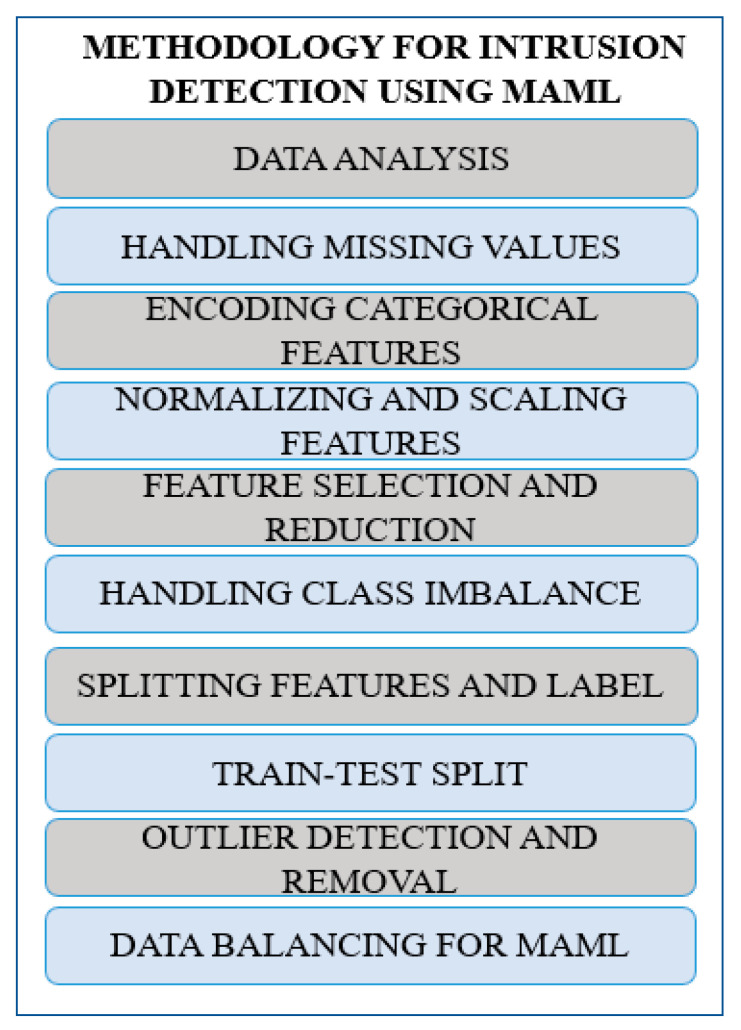
Methodology for UNSW-NB15 intrusion detection using MAML.

**Figure 6 sensors-25-02487-f006:**
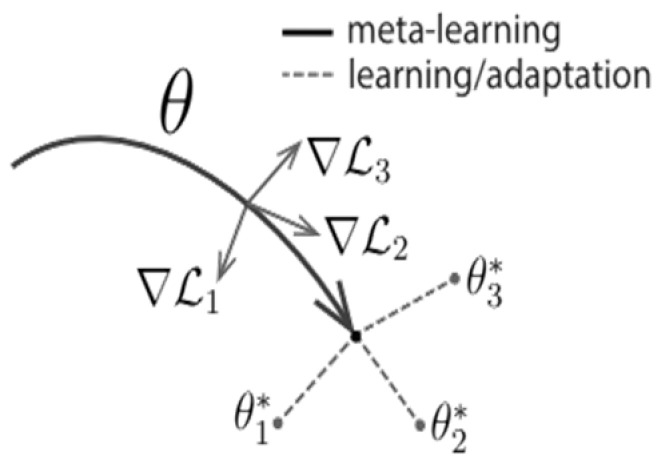
MAML learning adaption for theta, which can adapt to new tasks [37,38].

**Figure 7 sensors-25-02487-f007:**
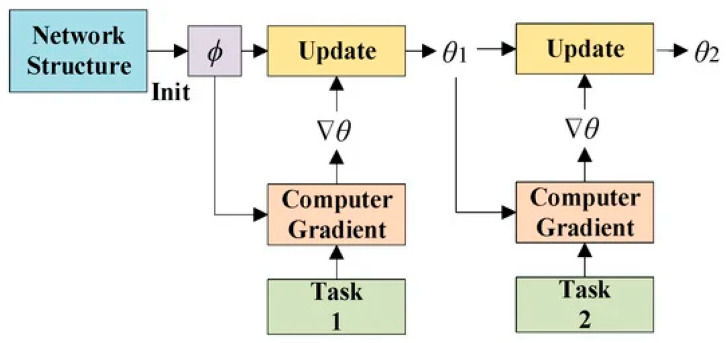
MAML workflow to update gradient.

**Figure 8 sensors-25-02487-f008:**
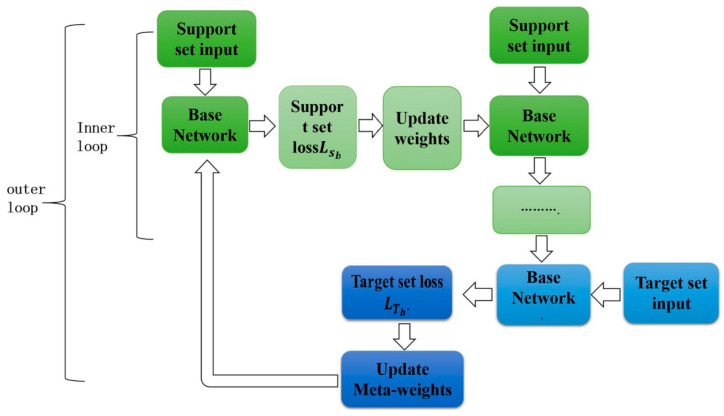
MAML workflow to update gradients and optimize performance.

**Figure 9 sensors-25-02487-f009:**
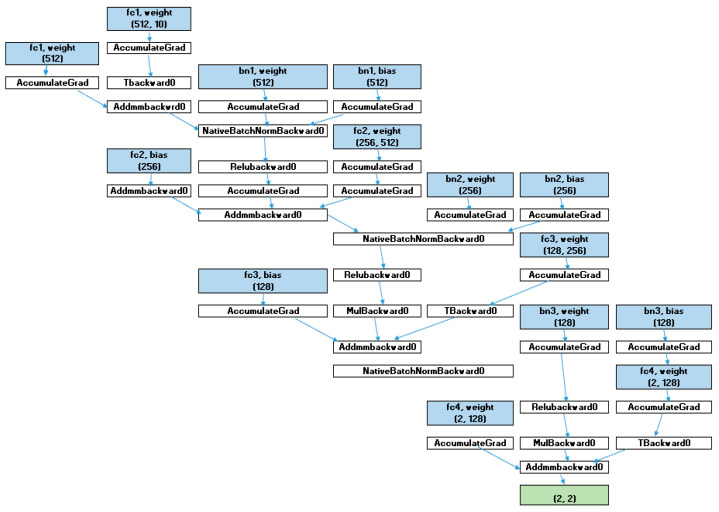
Proposed MAML layer distribution.

**Figure 10 sensors-25-02487-f010:**
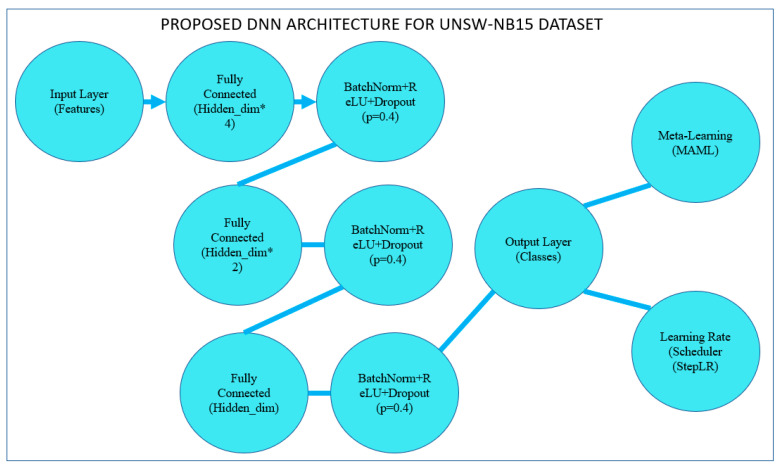
Proposed MAML architecture.

**Figure 11 sensors-25-02487-f011:**
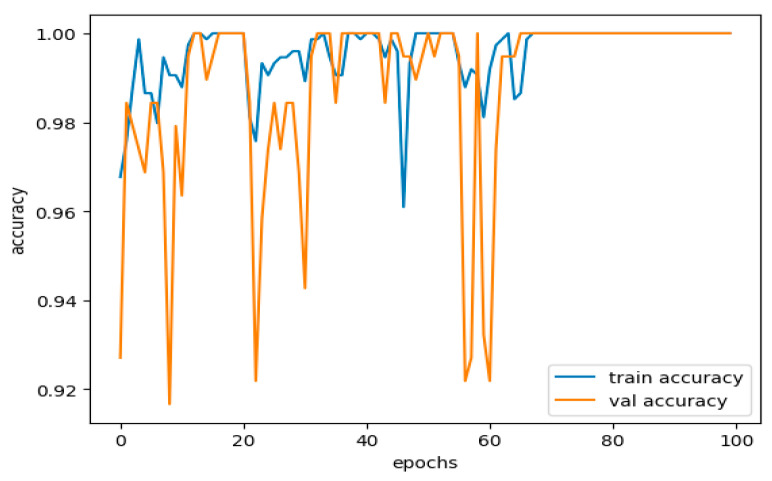
Training and validation accuracy performance curve.

**Figure 12 sensors-25-02487-f012:**
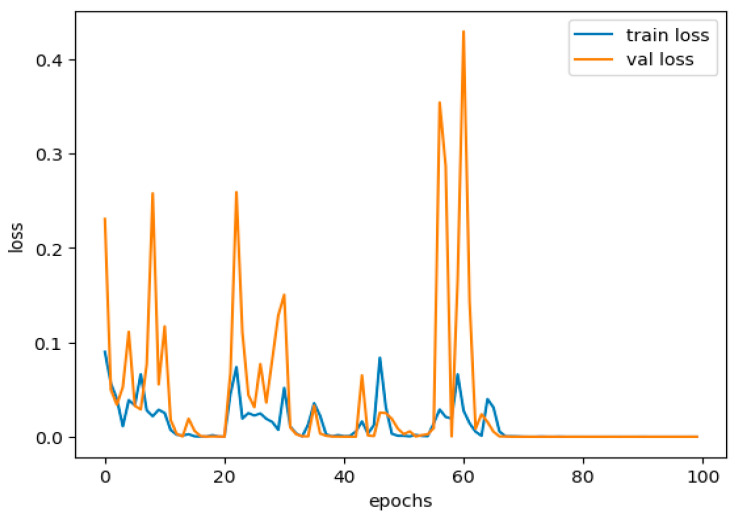
Training and validation loss performance curve.

**Figure 13 sensors-25-02487-f013:**
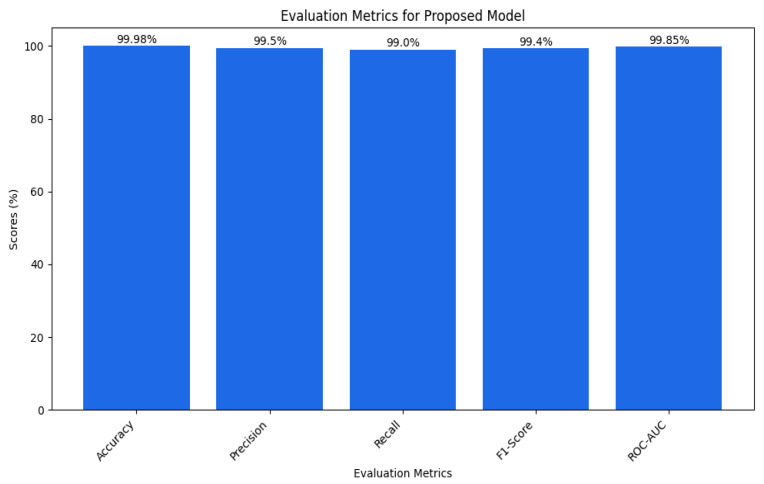
Evaluation metrics bar plot.

**Figure 14 sensors-25-02487-f014:**
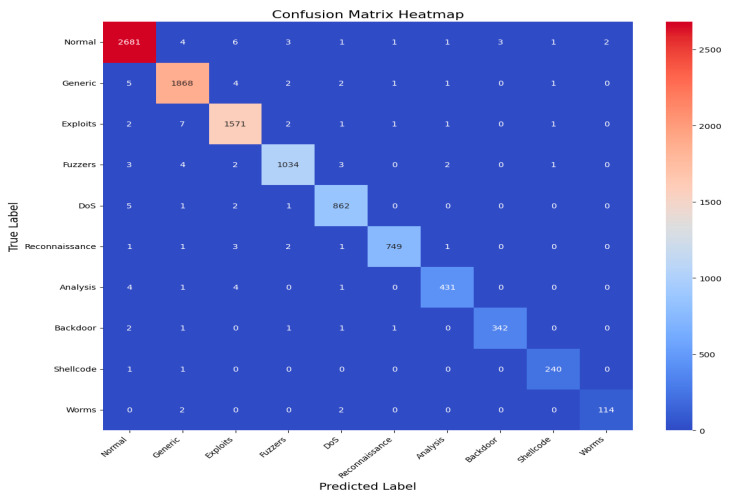
UNSW-NB15 confusion matrix across all classes.

**Figure 15 sensors-25-02487-f015:**
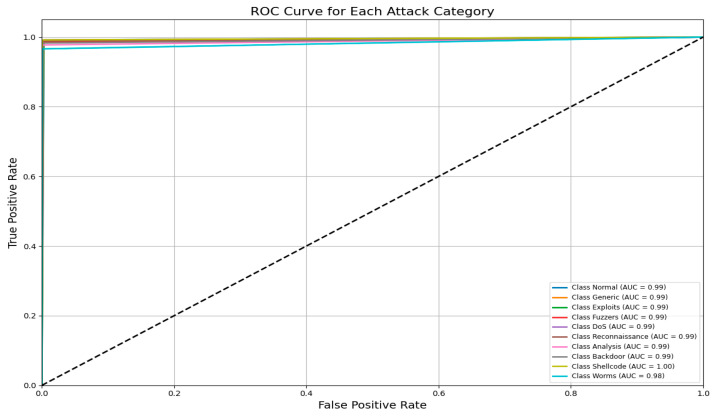
AUC–ROC curve for UNSW-NB15 across all classes.

**Figure 16 sensors-25-02487-f016:**
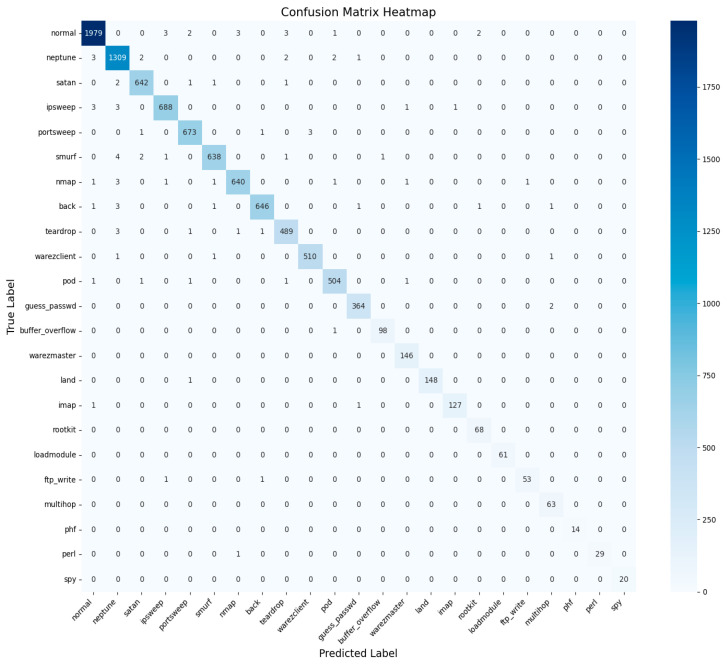
NSL-KDD99 confusion matrix across all classes.

**Figure 17 sensors-25-02487-f017:**
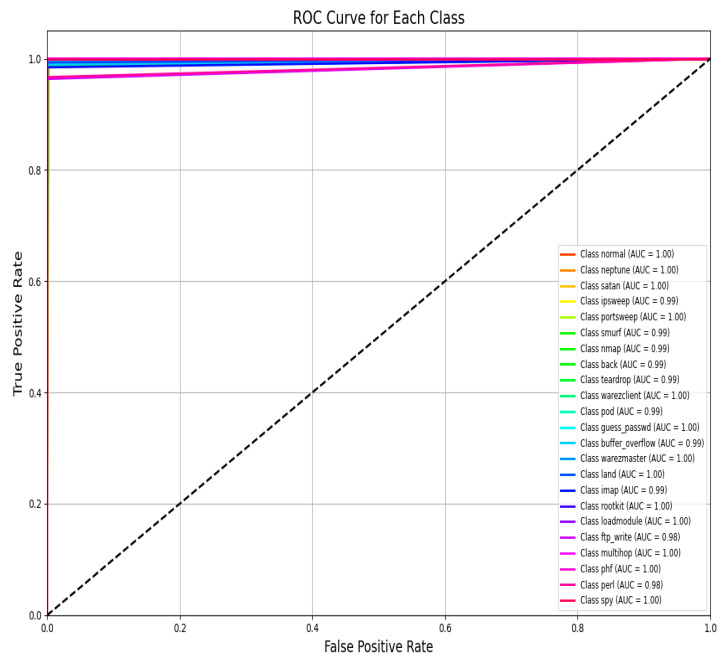
NSL-KDD99 AUC–ROC curve across all classes.

**Figure 18 sensors-25-02487-f018:**
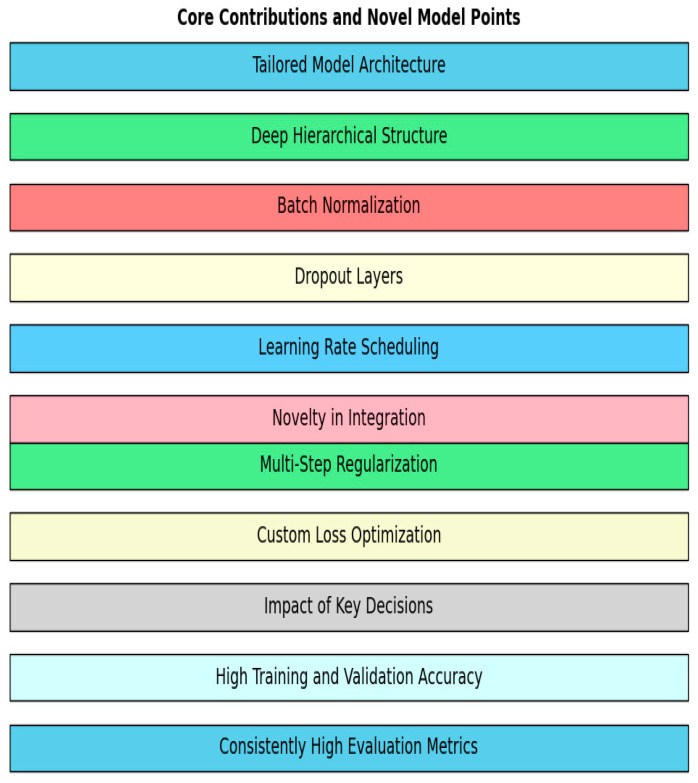
Core contributions and model novel design.

**Figure 19 sensors-25-02487-f019:**
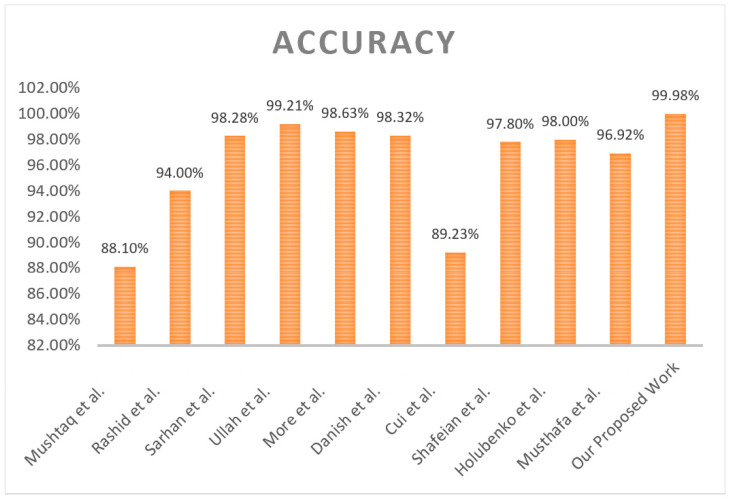
Comparative study performance [18,19,20,21,22,23,24,25,26,27].

**Table 1 sensors-25-02487-t001:** List of previous references, including datasets, methodologies, limitations, and results.

References	Datasets	Methodology	Limitations	Results
[16]	Limited network traffic samples for intrusion detection	Metric-based meta-learning and Adaptive Feature Fusion	Challenges in model information acquisition with FSL.	97.78% accuracy in multi-class few-shot tasks
[17]	Benign network traffic used for intrusion detection model training	Host-based anomaly detection using packet representations for security.	It relies solely on benign traffic and is limited to anomaly detection.	Accuracy 0.9874, precision 0.9384, recall 0.9971, F1 score 0.9529.
[18]	NSL-KDD benchmark dataset for intrusion detection evaluation.	Stacked ensemble with multiple classifiers and MLP.	A single classifier is insufficient for metamorphic malware detection.	88.10% accuracy, 0.87 detection rate, 0.17 false alarm.
[19]	NSL-KDD and UNSW-NB15 intrusion detection datasets	Tree-based stacking ensemble with feature selection techniques.	Ensemble methods require additional computation and complexity.	UNSW-NB15 accuracy of 0.9400 with XGBoost.
[20]	CSE-CIC-IDS2018, UNSW-NB15, and ToN-IoT for evaluation and testing.	ML models with PCA, AE, and LDA for feature extraction.	Performance varies, no universal feature set exists, and LDA is ineffective.	AE with 10 dimensions: DR 98.28%, FAR 3.21%
[21]	UNSW-NB15, CIC-IDS2017, NSL-KDD used for performance evaluation.	Transformer-based transfer learning with CNN-LSTM for attack detection.	Imbalanced data, complex features, and minority attack identification challenges.	Accuracy 99.21%.
[22]	UNSW-NB15 network traffic dataset used for cyber-attack detection.	LR, SVM, DT, and Random Forest algorithms were applied.	False positives and accuracy improvements are needed for IDS systems.	Random Forest: F1 97.80%, Accuracy 98.63%, FAR 1.36%.
[23]	CICIDS2018, ToN-IoT, Edge-IIoTset used for performance evaluation.	Hybrid approach: Bi-GRU, LSTM with softmax, TBPTT for learning.	Handling large data volumes, extreme environments, and attack exploitation.	Accuracy: 98.32%; FPR: 0.0426%.
[24]	UNSW-NB15 was used for model evaluation and testing.	Temporal convolutional residual modules with attention mechanism for detection.	Inadequate feature extraction and insufficient model generalization in prior methods.	Accuracy: UNSW-NB15 89.23%
[25]	Malicious data exfiltration and benign network traffic examples.	Comparison of bagging, boosting, and multi-layer stacking.	Few models meet strict acceptance criteria for detection.	0.978 accuracy, 0.001 false positive rate with MLP.
[26]	ADFA-LD dataset used.	Proposes lightweight Host-based Intrusion Detection System (HIDS) using system call traces and ML.	Limited focus on real-world implementation and HIDS research gap.	Achieved 98% accuracy and explained results using eXplainable AI.
[27]	NSL-KD and UNSW-NB15 datasets were used for performance evaluation and testing.	Class balancing, feature selection, SVM-bagging, LSTM-stacking with ANOVA.	Overfitting and feature selection challenges in Complex model optimization.	LSTM-stacking achieved 96.92–99.77% accuracy, low overfitting, high AUC.

**Table 2 sensors-25-02487-t002:** Computational parameters.

Metrics	Value
Training Time	40 min
Model Employed	MAML
Hardware Employed	Google Colabs GPU
Batch Size	32
Total Parameters	1.6 million
Epochs	100

**Table 3 sensors-25-02487-t003:** UNSW-NB15 evaluation metrics.

Evaluation Metric	Value
Accuracy	99.98
Recall	99.0
Precision	99.5
F1 score	99.4

**Table 4 sensors-25-02487-t004:** NSL-KDD99 evaluation metrics.

Evaluation Metric	Value
Accuracy	99.1
Recall	98.2
Precision	97.3
F1 score	98.5

**Table 5 sensors-25-02487-t005:** Comparative studies for intrusion detection.

References	Methodology	Accuracy	Dataset
[18]	Stacked ensemble	88.10%	UNSW-NB15
[19]	Tree-based stacking ensemble	94%	UNSW-NB15
[20]	ML models with PCA, AE, LDA	98.28%	UNSW-NB15
[21]	Transformer-based transfer learning	99.21%	UNSW-NB15
[22]	LR, SVM, DT, RF	98.63%	UNSW-NB15
[23]	Hybrid approach: Bi-GRU, LSTM	98.32%	UNSW-NB15
[24]	Temporal convolutional residual modules	89.23%	UNSW-NB15
[25]	Bagging, boosting, multi-layer stacking	0.978	UNSW-NB15
[26]	Lightweight HIDS, ML	98%	UNSW-NB15
[27]	Class balancing, SVM-bagging, LSTM-stacking	96.92%	UNSW-NB15
Our Proposed Work	MAML	99.98% for UNSW-NB15 and 99.1% for NSL-KDD	UNSW-NB15 NSL-KDD

## Data Availability

Dataset is available on reasonable request.

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
