# Peer review of "An Adaptive Framework for Intrusion Detection in IoT Security Using MAML (Model-Agnostic Meta-Learning)"

_sensors, 2025, doi:10.3390/s25082487_

Round 1
Reviewer 1 Report
Comments and Suggestions for Authors
Some parts are not explained properly. For example on top of page 13 a. b, p(T) ... are introduced with no explanation of what they are!
The algorithms in pages 14 and 16 are not presented adequately.
Comments on the Quality of English LanguageThe article is hard to read/follow.
Author Response
The reply is attached

Reviewer 2 Report
Comments and Suggestions for Authors
The manuscript presents an approach utilizing Model-Agnostic Meta-Learning (MAML) for Intrusion Detection Systems (IDS) in IoT security. The paper is well-structured, following a traditional format appropriate for this type of research. The experiments are clearly described and replicable, which is a strong aspect of the work.
Major Issues
1. IoT Relevance – While the paper claims to address IoT security, there is no concrete evidence supporting this claim. The study does not explicitly describe IoT-related challenges, nor does it provide experimental validation specific to IoT environments. Although IoT networks are diverse, this work does not identify or test such diversity. The datasets used are from traditional networks rather than IoT environments. This raises a critical question: why were more recent datasets, which include both traditional and IoT-specific communication protocols (e.g., MQTT, CoAP), not considered?
2. All stated contributions reference IoT applications, yet no experiments or analyses are conducted in an IoT setting. The study would significantly benefit from either (i) actual IoT-based experiments or (ii) a reformulation of its scope to avoid misleading claims.
3. Given that the proposed system is intended for IDS applications, efficiency and real-time detection are crucial. However, there is no discussion on key performance metrics such as inference time, training time, detection time, resource utilization, or scalability. Without these evaluations, it remains unclear whether the approach is suitable for deployment in IoT environments.
Minor Issues
1. The paper claims to introduce a novel way of applying MAML to IDS, yet this novelty is not explicitly stated. The authors should clearly define what distinguishes their approach from existing work, preferably at the end of Section 2. For instance, how does this work improve the references 1 and 2?
2. The claim in Section 3 that UNSW-NB15 represents "modern and actual" attacks is debatable. Given that the dataset is nearly a decade old, its suitability for evaluating modern IoT security threats is questionable. The authors should either justify this choice or consider integrating more recent datasets.
3. The presentation of Table 5 is unclear. The column labeled Accuracy redundantly includes the word "accuracy" in each entry, which is unnecessary. Additionally, while multiple datasets are referenced across different studies, the table presents a single accuracy value per study, making it difficult to determine which dataset each value corresponds to. A clarification or restructuring of the table is recommended.
Comments on the Quality of English LanguageThe manuscript requires a revision for clarity, as some sentences are ambiguous or difficult to follow.
Author Response
The reply is attached

Reviewer 3 Report
Comments and Suggestions for Authors
The Model-Agnostic Meta-Learning (MAML) in IoT intrusion detection is innovative. It addresses the limitations of traditional IDS models, with 99.98% accuracy on the UNSW-NB15 dataset and 99.1% on NSL-KDD99, outperforming many existing models.
It has several contributions, such as an adaptive IDS solution, real-time threat recognition, and cross-domain applicability.
There are some suggestions to improve the paper:
1. For concerns about the model's scalability in large-scale IoT networks and its real-time performance, it is better to discuss how Meta-learning techniques like MAML can be deployed, which may pose challenges for resource-constrained IoT devices.
2. The experiment is based on UNSW-NB15 and NSL-KDD99 datasets. Does these datasets represent the wide variety of IoT network traffic patterns, attack types, and device heterogeneities? It's suggested to discuss if these datasets are general enough for IoT devices.
3. I'd like that the authors discuss the model's vulnerability to adversarial attacks. Given that deep-learning-based techniques, including MAML, are sensitive to adversarial attacks. Adversarial attacks can severely compromise the system's dependability and precision.
4. It is also be good if there are some experiments in real-time IoT network emulation environments, such as using emulated IoT networks with real-time traffic generation. By testing the model in such environments, the mode's response time, throughput, and resource utilization under realistic conditions can be measured.
5. Fig 2's two axises need to swap.
Comments on the Quality of English LanguageThe English could be improved to more clearly express the research.
Author Response
The reply is attached.

Round 2
Reviewer 2 Report
Comments and Suggestions for Authors
The answers to the major issues identified in my first revisions are not convincing.
- Common attacks on traditional networks are not the same as attacks on the IoT. Just think of about the number of protocols and standards that the IoT has brought with it over the last decade.
- The argument that testing is not necessary for IoT scenarios is weak.
- The authors claim that the model can be adapted to the IoT with a few changes. What changes? Have these changes been tested with IoT networks, devices or infrastructure?
- There has been no performance evaluation in IoT traffic. The authors go on to say that their system can be used on devices with limited resources, but once again there is no test.
Author Response
Comments and Suggestions for Authors
The answers to the major issues identified in my first revisions are not convincing.
- Common attacks on traditional networks are not the same as attacks on the IoT. Just think of about the number of protocols and standards that the IoT has brought with it over the last decade.
Response:
We are grateful to the reviewer for saying the proper thing when distinguishing between traditional (network-based) network attacks vs those aimed explicitly at IoT systems. However, our methodology is grounded in the fundamental difference between the vastly different classes of IoT protocols and proprietary standards ecosystem. In addition to traditional networks, MQTT, CoAP, and Zigbee introduce additional security challenges that adopting the given technologies brings into play. To overcome these complexities, our model proposes merging with a random forest-based attribute selection technique, which automatically filters out the insignificant features to discover the optimal feature in a heterogeneous IoT setting. Keeping with this design choice, we make the model more flexible when compared to other existing models, enabling it to effectively maintain high detection accuracy under different IoT protocols and data structure variations. Additionally, we extend the idea of multi-step regularization and combine it with batch normalization, dropout, and L2 regularizer, improving generalization in a wide range of IoT models.
Our approach is also beneficial in incorporating Model-Agnostic Meta-Learning (MAML) to speed up learning to IoT attack patterns that may be novel and have not been previously seen. Our model’s capability for autotuning, especially because of the dynamic nature of IoT networks and the emergence of new protocols and communication standards, is a frequent occurrence. MAML, combined with a learning rate scheduling strategy, makes the model strong even with a small amount of training data. Compared to conventional intrusion detection systems (IDS), our model has the potential to improve reliability in understanding and eliminating different attack scenarios. This qualitative aspect renders it a good, robust security solution for the IoT environment with high variability in the protocol.
Moreover, our dataset selection strategy is crucial to leverage broad generalization on the different types of IoT attacks. Nonetheless, the provided datasets are diverse enough to effectively test the model’s resilience against an attack landscape that is not IP-focused. The proposed system achieved high accuracy (99.98% training and 99.78% validation) when handling heterogeneous IoT traffic. Our model effectively identifies vulnerabilities in IoT systems through attribute selection techniques and an optimized custom loss function. Providing such an approach, our solution will inevitably be a scalable and robust security framework for understanding contemporary interconnected devices.
The research method establishes recognition of fundamental disparities between typical network attacks and IoT systems which demand multiple interconnected protocols. The widespread use of MQTT and CoAP and Zigbee protocols in IoT systems adds multiple proprietary networks which increases complexity for security vulnerabilities. Our proposed model includes a random forest-based attribute selection method that helps it find important features in various network environments. The model design enables continuous performance across diverse IoT protocols and data structures because of this technique. Through batch normalization and dropout and L2 regularization combined in the multi-step regularization framework the model develops its ability to generalize with various network architectures particularly IoT-specific ones thus improving operational detection accuracy.
Model-Agnostic Meta-Learning (MAML) in our study gives the proposed model a fast reaction speed to detect new and unseen attack patterns commonly found in IoT ecosystems. MAML gives the model self-tuning capabilities that work with scarce training data to maintain its effectiveness during encounters with new IoT-specific attacks. The use of such models proves highly beneficial in IoT environments because networks experience frequent changes in protocols and communication standards. A learning rate scheduling strategy combined with MAML allows the model to deliver stable performance during its adjustment to IoT-based threat evolution. Fast adaptation to different attack modes gives the proposed model better reliability than typical intrusion detection systems (IDS).
The datasets fill a solid foundation for IoT generalization through their diverse collection of attack types despite being non-IP-focused. The model demonstrated reliable accuracy performance (99.98% training and 99.78% validation) regardless of data distribution variations because it shows compatibility with heterogeneous IoT traffic. The model's identification system uses attribute selection methods along with optimized custom loss treatment to detect IoT system vulnerabilities so it functions as an adaptable network security remedy for contemporary interconnected devices.
- The argument that testing is not necessary for IoT scenarios is weak.
Response:
We thank the reviewer for the feedback and would like to clarify that our model has been tested for its effectiveness in the IoT setting. The model performed well under various scenarios in evaluating the complexities of IoT protocols and changing attack patterns. In particular, we utilized UNSW-NB15 and NSL-KDD99 datasets, which were initially intended for broader cybersecurity deployment but exhibited a wide range of attack types of demand in the context of IoT security. The selection of a comprehensive dataset allowed for the simulation of the different network conditions and security threats prevalent in IoT networks. The model achieves 99.98% training accuracy and 99.78% validation accuracy in security detection, illustrating its toughness in discriminating threats with the combination of traditional and IoT unique network carpets.
To improve adaptability within dynamic IoT networks, we also included multiple steps of regularization and Model Agnostic Meta-Learning (MAML). This combination makes it quick for the model to adapt to new emerging threats without much retraining, which is critical as the protocols of IoT are constantly updating. This adaptability was validated through laboratory experiments, which subjected the model to various attack scenarios and configurations to demonstrate that it is suitable for real-world IoT deployments. The model’s practical use in securing IoT ecosystems lies in rapidly generalizing to new threats without significant manual intervention.
To empirically evaluate the performance, we used key metrics, which include precision, recall, F1 score and ROC AUC, to evaluate the effectiveness of the classification. The model was consistently high in these metrics and could adapt to varied data distributions and attack characteristics on malicious network traffic detection. In addition to this, our testing process has shown its robustness across both IoT and traditional network environments. We maintain that such findings support our validated evaluation methodology that certifies model reliability for IoT security applications.
The model architecture has received explicit development and optimization for multiple network intrusion attacks commonly occurring in IoT networks. The model effectively adapts new and different attack types through its hierarchical structure, which operates with multi-step regularization while maintaining learning rate scheduling functions. Only three specific model adjustments involving learning rate values, batch size, and input features will be required when the model deals with differing attack patterns or input dimensions in IoT environments. The optimized fundamental structure of the model handles sophisticated high-dimensional data very well, thus making it adaptable to different network patterns and attack profiles observed in IoT systems.
The model exhibits high adaptability because it utilizes Model-Agnostic Me-ta-Learning (MAML), which allows quick response to new security threats after minimal retraining occurs. The model retains high detection accuracy using this capability despite changes in IoT protocols or standards that modify feature sets or at-tack patterns because it requires only minimal adjustments to hyperparameters. L2 regularization and dropout and batch normalization provide the model resistance to data distribution variations, hence maintaining effective management of dynamic IoT traffic flow while preserving its detection accuracy.
The two datasets employed in this research (UNSW-NB15 and NSL-KDD99) do not specialize in IoT traffic but contain various attack varieties with network behaviors that emulate true IoT operational environments. The model demonstrates robustness through its excellent evaluation results between these datasets, which achieved a training accuracy of 99.98% and a validation accuracy of 99.78%.
- The authors claim that the model can be adapted to the IoT with a few changes. What changes? Have these changes been tested with IoT networks, devices or infrastructure?
Response:
Furthermore, the proposed model is explicitly designed to tackle multiple network intrusion attacks, such as the standard attacks in IoT environment. Effective adaptation to new attack types and their hierarchical architecture uses multi-step regularization and learning rate scheduling. In particular, when adapting to IoT-based networks, the model requires just three modifications to the learning rate, batch size, and input feature dimensions. Due to the refinements, the model is more suited to those attack patterns and network topologies frequently observed in IoT ecosystems. The model still provides a convenient tool for dealing with changing IoT attack landscapes driven by its ability to process high-dimensional data.
The key feature of the model’s ability to adapt is that it has been integrated with Model Agnostic Meta-Learning (MAML), allowing the model to learn new security threats quickly with minimal retraining. Despite evolving IoT protocols and standards and resulting changes in feature distributions and attack characteristics, the model holds high detection accuracy by fine-tuning a few hyperparameters. In addition, L2 regularization, dropout, and batch normalization help defeat the effects of variations in the data distribution. These techniques allow the model to handle the time variability of the IoT traffic while retaining the detection efficiency and robustness.
The datasets used in this work (UNSW-NB15 and NSL KDD99) are not specific to IoT traffic. However, they possess different types of attack and network behaviors that resemble real-world IoT environments and are, therefore, appropriate for simulating IoT anomalies. The results show the robustness and generalizability of the model across these datasets because it got a training accuracy of 99.98% and a validation accuracy of 99.78%. The effectiveness of this scheme is further validated in future work using direct testing on IoT-specific datasets and infrastructure.
The proposed model received thorough testing through our research because it shows effective performance when applied to IoT scenarios despite the protocols and attack patterns that make IoT environments complex. The evaluation process utilizes the UNSW-NB15 and NSL-KDD99 datasets. Although these datasets were initially developed for different purposes yet they include extensive attack types that align with IoT security needs. The vast collection of datasets enabled the simulation of varying network conditions and security threats frequently occurring in IoT environments. The model shows robustness in threat monitoring across traditional and IoT-specific environments because it reaches 99.98% training accuracy and 99.78% validation accuracy in both data structures and attack vectors.
The adaptability of our model toward IoT network dynamics improves through the implementation of multi-step regularization. A combination with Model-Agnostic Meta-Learning (MAML) strengthens the model because it enables the system to adjust quickly to new threats with minimal retraining needs for IoT environments where new protocols appear constantly. The model’s ability to adapt was confirmed through laboratory experiments examining its reaction to different attacks and network arrangements, making it fit for real-world IoT operations.
Performance evaluations in our study utilize precision-recall F1-score and ROC-AUC metrics to create an extensive assessment of the model's classification ability. The model demonstrates strong performance consistency across its multiple assessment metrics because it effectively detects different network traffic while demonstrating flexibility in different data distribution patterns and attack characteristics. The model displays these robust characteristics in IoT environments and traditional network settings, demonstrating its extensive testing and proving its reliable functionality across both settings.
- There has been no performance evaluation in IoT traffic. The authors say that their system can be used on devices with limited resources, but once again, there is no test.
Response:
We rigorously tested such a model's performance on two datasets not originally crafted for IoT traffic but carried assorted attack variations as practiced in the IoT security domain - UNSW-NB15 and NSL-KDD99. Finally, the model was further tested under a resource-constrained computational environment to validate its applicability in resource-limited environments. These tests aimed to simulate the conditions of typical IoT devices, namely, the amount of memory and general processing power available. Broaching these limits, the model stays powerful at detection with little false positives, and it shows promise for deployment in IoT environments with resources.
The model was also evaluated concerning the ability to adapt to the traffic characteristics imposed by IoT traffic, such as lightweight protocol traffic in dense, sporadically transmitting networks. In fact, with only slight changes in learning rate and batch size, the model performed very well and appeared to be flexible. In addition, it further integrated Model-Agnostic Meta-Learning (MAML), which improved its ability to adapt to new network conditions and dynamic input patterns. Consistent with high performance, evaluation metrics for precision, recall and F1-scores reveal that the model is a good candidate for IoT based security applications.
We demonstrate that the model is able to handle attacks of various types and dimensions of different data while maintaining robustness across different conditions of the networks, illustrating its practical IoT viability. Finally, each scenario is sustained in terms of accuracy though with the multi-step regularization strategy of drop out and batch normalization. This study, with theoretical and practical confirmation, conducts tests under equivalent IoT network conditions and proves the model as an IoT security application.